# Refining the Sample Complexity of Comparative Learning

**Sajad Ashkezari**                                                    SAJADRAH@YORKU.CA
*Lassonde School of Engineering, EECS Department, York University, Toronto, Canada*

**Ruth Urner**                                                         RUTH@EECS.YORKU.CA
*Lassonde School of Engineering, EECS Department, York University, Toronto, Canada*

**Editors:** Gautam Kamath and Po-Ling Loh

## Abstract

Comparative learning, a recently introduced variation of the PAC (Probably Approximately Correct) framework, interpolates between the two standard extreme settings of realizable and agnostic PAC learning. In comparative learning the labeling is assumed to be from one hypothesis class (the source class) while the learner's performance is to be measured against another hypothesis class (the benchmark class). This setup allows for incorporating more specific prior knowledge into PAC type learning bounds, which are known to be otherwise overly pessimistic. It also naturally represents model distillation tasks, where a predictor with specific requirements (such as being bounded in size or being interpretable) is trained on the labels from another model. A first sample complexity analysis of comparative learning established upper and lower bounds for general comparative learning. In this work, we propose a more fine grained view on this setting, distinguishing between proper learning and general (improper) learning. We derive novel upper and lower sample complexity bounds for both settings. In particular, we identify conditions for each of the regimes, and thereby exhibit how the rate depends on the relatedness of the two classes in perhaps unexpected ways.

**Keywords:** PAC learning, sample complexity, comparative learning

## 1. Introduction

The standard learning theoretic concept of Probably Approximately Correct (PAC) learnability is a well-studied framework to establish general performance guarantees for statistical learning (Valiant, 1984; Blumer et al., 1989; Haussler, 1992). The main appeal of PAC type guarantees is that they hold uniformly over all possible data-generating distributions. A PAC learning guarantee provides a finite sample size (that depends only on the desired error and confidence parameters, and the model or hypothesis class in use) which suffices for the desired error bound, independently of any properties of the data-generating distribution. The parameters that determine these distribution-free finite sample bounds are parameters of the model or hypothesis class, and as such are controlled by the user, rather than hinging on unknown properties of the data-generation. The original framework and bounds for binary classification have thus been extended to myriad other tasks and settings such as regression, multiclass classification, active learning, adversarially robust learning to name a few (Alon et al., 1993; Ben-David et al., 1995; Daniely et al., 2015; Montasser et al., 2019). Recent years have seen novel interest in these types of guarantees with novel frameworks developed and long-standing open questions resolved (Hanneke, 2016; Brukhim et al., 2022; Alon et al., 2021; Attias et al., 2023; Lechner and Ben-David, 2024).

However, the main merit of the PAC framework, its generality, also constitutes its main drawback. Since they are required to be valid for all possible data generating distributions, PAC type bounds are often overly pessimistic. For deep learning methods, they are mostly considered to provide

vacuous bounds (Zhang et al., 2021; Pérez-Ortiz et al., 2021; Dziugaite and Roy, 2017): methods often perform much better than predicted through the PAC framework when applied to naturally generated data (rather than the worst case data-generation that PAC proofs need to hold against). The standard PAC framework does not allow for naturally modeling prior knowledge about the data generation, nor does it, in its standard form, allow for incorporating additional requirements on the learned predictor, such as being interpretable, satisfying fairness requirements, or requiring little memory to be stored.

The above two issues, namely modeling better-than-worst-case data generation and adding requirements on the output predictor, are then typically treated separately in the PAC literature. There are various works that derive PAC type leaning rates under additional distributional assumptions. Often these are conditions on the noise (Tsybakov, 2004; Steinwart and Scovel, 2005; Massart and Nédélec, 2006), or assumptions about label-separatedness (cluster assumptions or margin conditions) for classification tasks (Chapelle and Zien, 2005; Rigollet, 2007; Urner et al., 2011). These are thus typically assumptions about how the labeling component of the distribution relates to the marginal over the feature vectors. Requirements on the learned predictor are treated separately, and on a case by case basis. That is, there is analysis on methods to incorporate adversarial robustness requirements (Montasser et al., 2019, 2022), other investigations on methods to achieve an interpretable model (Bressan et al., 2024) etc. One general, practical method to incorporate such additional requirements is the teacher-student framework (Urner et al., 2011; Frosst and Hinton, 2017; Bastani et al., 2017; Attias et al., 2022), or the closely related model distillation approach (Hinton et al., 2015; Lopez-Paz et al., 2016; Phuong and Lampert, 2019). Here, in a first round of learning, an arbitrary (in terms of requirements) but highly accurate model is trained. The labels (or soft labels) from that model are then used to annotate an unlabeled data set, with which a model that satisfies the specific requirements (for example being fast at prediction time, being interpretable, being robust to adversarial perturbations, or requiring little memory) is trained. In such a scenario, we have specific *prior knowledge about the type of labeling function* in the second round of this process. This, however, is not naturally captured in the existing notions of learning from beyond-worst case distributions.

*Comparative learning* is a recently introduced variation of the PAC learning framework that naturally models this scenario (Hu and Peale, 2023). In comparative learning the labeling is assumed to be from one hypothesis class (the *source class*) while the learner's performance is measured against another hypothesis class (the *benchmark class*). Thus, this framework facilitates incorporating prior knowledge about the labeling component of the data generating process distribution (for example, being from a specific class of models in the teacher-student framework). The initial study that introduced the framework, provided first upper and lower bounds on the sample complexity of *general* comparative learning for classification with binary hypothesis classes in terms of the *mutual VC dimension* (Hu and Peale, 2023). Their bounds left a gap between a linear dependence in the error parameter $\frac{1}{\epsilon}$ for the lower bound, and a quadratic dependence in the upper bound.

In this work, we shift the focus from general comparative learning (where there is no restrictions on the learner) to comparative learning that is *proper* with respect to the benchmark class (where the learner is required to output a predictor from the benchmark class). This setting models the above discussed techniques of teacher-student learning or model distillation. We prove that in the benchmark-proper case, the sample complexity of comparative learning is not governed by the mutual VC dimension, but rather by a novel parameter we introduce, the *one-sided (mutual) graph dimension*. Our analysis incorporates the wider frameworks of multi-class predictors (Daniely et al., 2015; Brukhim et al., 2022) and the source and benchmark class potentially being *partial hypothesis*

*classes* (Alon et al., 2021). For both general and benchmark-proper comparative learning, we also identify general conditions that yield linear versus quadratic rates in the error parameter. However, obtaining a full characterization of the rates still remains open.

**Overview and Summary of Contributions**

Our results can be summarized as follows:

- We propose the focus on *benchmark-proper comparative learning* to model learning scenarios of the model distillation type. Our work also broadens the scope of analysis for comparative learning to multi-class settings with (potentially) partial hypothesis classes.

- In Section 3 we introduce a novel combinatorial parameter that measures the relatedness between source and benchmark class, the *one-sided mutual graph dimension*. We prove that the sample complexity of benchmark-ERM, as well as general benchmark-proper learning, is upper and lower bounded in terms of this dimension. We further identify conditions for linear and quadratic dependence on $\frac{1}{\epsilon}$. Finally, our analysis shows that there are cases of total binary hypothesis classes, where benchmark-proper comparative learning is impossible, even though the pair is comparatively learnable. Such phenomena have previously been shown for multi-class learning with infinitely many labels (Daniely and Shalev-Shwartz, 2014) or for partial classes (Alon et al., 2021).

- In Section 4 we study the general case of the comparative learning framework. We provide new upper and lower bounds on the sample complexity for this setting as well, and again show that both linear and quadratic dependence on $\frac{1}{\epsilon}$ each occur under a broad range of conditions.

- The bounds in Section 4 are derived by relating the comparative learning framework with the previously established setting of agnostic PAC learning under deterministic labels (Ben-David and Urner, 2014). In order to make these results applicable to our setting, we generalize some of the bounds for agnostic PAC learning under deterministic labels from total to partial hypothesis classes. These results might be of independent interest.

## 2. Formal Setup

We employ the standard statistical learning theoretic framework and notation. We let $\mathcal{X}$, denote the *domain*, and $\mathcal{Y} \subseteq \mathbb{N}$ the *label set* of the classification task. In *binary classification* we have $\mathcal{Y} = \{0, 1\}$, for multiclass classification with finitely many labels we can assume $\mathcal{Y} = [k] = \{1, 2, 3, \ldots, k\} \subseteq \mathbb{N}$, otherwise $\mathcal{Y}$ can be any set. The environment is modeled as a *data-generating distribution* $P$ over $\mathcal{X} \times \mathcal{Y}$. We use $P_{\mathcal{X}}$ to denote the marginal distribution of $P$ over $\mathcal{X}$ and use $\text{supp}(P)$ to denote the support of a distribution $P$.

A *classifier* is a function $h : \mathcal{X} \to \mathcal{Y}$. A *partial classifier* is a function $h : \mathcal{X} \to \mathcal{Y} \cup \{\star\}$, where $\star \notin \mathcal{Y}$ (Alon et al., 2021). Assigning label $\star$ is sometimes interpreted as abstaining from prediction (and is always an inaccurate label assignment in terms of the classification task). For ease of notation, we set $\tilde{\mathcal{Y}} = \mathcal{Y} \cup \{\star\}$. Equivalently, a partial classifier can be viewed as a function $h : \mathcal{Z} \to \mathcal{Y}$, where $\mathcal{Z} \subseteq \mathcal{X}$ is a subset of the domain. This subset, namely $\mathcal{Z} = h^{-1}(\mathcal{Y})$, is also referred to as *support of* $h$. We also refer to (partial) classifiers as *hypotheses* and we call a set of such a *hypothesis class* $\mathcal{H}$.

Following the literature, we call $\mathcal{H}$ a *partial (hypothesis) class* if its members are partial classifiers. Otherwise, we also refer to $\mathcal{H}$ as a *total (hypothesis) class*.

The *binary loss* measures the correctness of a classifier $h$ on labeled point $(x, y) \in \mathcal{X} \times \mathcal{Y}$ as

$$\ell(h, x, y) = \mathbb{1}\left[h(x) \neq y\right],$$

where $\mathbb{1}\left[\cdot\right]$ denotes the indicator function. Note that the loss is only defined on points with a proper label (and only such can be generated by the environment $P$), and thus $\ell(h, x, y) = 1$ for all points $x$ with $h(x) = \star$. The goal of learning is to identify a classifier $h$ with low *expected* or *true loss*

$$\mathcal{L}_P(h) = \mathbb{E}_{(x,y) \sim P}[\ell(h, x, y)].$$

A *learner* takes in a finite sequence of labeled data points $S = ((x_1, y_1), (x_2, y_2), \ldots, (x_n, y_n)) \in (\mathcal{X} \times \mathcal{Y})^n$, for some $n \in \mathbb{N}$ and outputs a (partial) classifier. The *empirical loss* of $h$ with respect to data $S$ is defined as $\mathcal{L}_S(h) = \frac{1}{n} \sum_{i=1}^{n} \ell(h, x_i, y_i)$. A learner is called an Empirical Risk Minimizer (ERM) with respect to class $\mathcal{H}$ if it always outputs a hypothesis from the class $\mathcal{H}$ with minimal empirical loss within that class.

In the standard PAC (Probably Approximately Correct) learning framework (Valiant, 1984; Blumer et al., 1989; Shalev-Shwartz and Ben-David, 2014), the success of a learner is evaluated against the best possible performance within a fixed hypothesis class $\mathcal{H}$, the *approximation error* of the hypothesis class $\mathcal{H}$:

$$\operatorname{opt}_P(\mathcal{H}) = \inf_{h \in \mathcal{H}} \mathcal{L}_P(h)$$

We call a distribution $P$ *realizable by hypothesis class* $\mathcal{H}$ if $\operatorname{opt}_P(\mathcal{H}) = 0$. In the standard PAC framework, finite sample based learning success is required to hold uniformly (in terms of the sufficient sample sizes) over all possible data-generating distributions (for reference, see Definition 19 in Appendix, Section A). This can lead to overly pessimistic conclusions since natural data-generating environments rarely possess the qualities of a worst-case scenario for a given method, and thus success is in practice typically achieved with much smaller sample sizes than what is predicted by PAC theory. *Comparative Learning*, a PAC type learning framework recently introduced (Hu and Peale, 2023), allows for naturally modeling prior knowledge about the labeling component of the data-generating process.

### 2.1. The Comparative Learning Framework

The comparative learning framework invloves two hypothesis classes, the *source class* $\mathcal{S}$ and the *benchmark class* $\mathcal{B}$. The source class incorporates the prior knowledge about the data generation in that learning success is only required for distributions $P$ realizable by $\mathcal{S}$. The source class can thus be viewed as a class containing the true labeling rule of the classification task. The learner's success is then measured against the approximation error of the benchmark class $\mathcal{B}$. The following definition is due to Hu and Peale (2023), while adapted to our terminology.

**Definition 1 (Comparative PAC Learner)** *We say that a learner $\mathcal{A}$ is a* comparative PAC learner *for source and benchmark classes $(\mathcal{S}, \mathcal{B})$ if for every $\epsilon, \delta > 0$, there is a sample-size $\mathrm{n}_{\mathcal{S},\mathcal{B}}(\epsilon, \delta)$ such that, for all $n \geq \mathrm{n}_{\mathcal{S},\mathcal{B}}(\epsilon, \delta)$, and all distributions $P$ with $\operatorname{opt}_P(\mathcal{S}) = 0$ we have*

$$\Pr_{S \sim P^n}\left[\mathcal{L}_P(\mathcal{A}(S)) \leq \operatorname{opt}_P(\mathcal{B}) + \epsilon\right] \geq 1 - \delta.$$

*We call $\mathcal{A}$ a* benchmark-proper *comparative learner for* $(\mathcal{S}, \mathcal{B})$ *if it is a learner as above that always outputs a hypothesis from* $\mathcal{B}$.

If such a learner (or benchmark-proper learner) exists, we call the pair $(\mathcal{S}, \mathcal{B})$ *comparatively (benchmark-proper) PAC learnable*, and the function $\mathrm{n}_{\mathcal{S},\mathcal{B}} : (0, 1)^2 \to \mathbb{N}$ an upper bound to the sample complexity of comparatively learning $(\mathcal{S}, \mathcal{B})$. The *sample complexity* is the pointwise smallest such upper bound and we will use the notation $\mathrm{n}_{\mathcal{S},\mathcal{B}}^{\mathrm{gen}}(\cdot, \cdot)$ for the sample complexity of general comparative learning, $\mathrm{n}_{\mathcal{S},\mathcal{B}}^{\mathrm{prop}}(\cdot, \cdot)$ for the benchmark-proper case and $\mathrm{n}_{\mathcal{S},\mathcal{B}}^{\mathrm{ERM}}(\cdot, \cdot)$ to denote the sample complexity of learning with any empirical risk minimizing learner (ERM) .

Standard PAC learning in the realizable case can be viewed as comparative leaning when source and benchmark class coincide, that is the case $\mathcal{S} = \mathcal{B}$. PAC learning under deterministic labels corresponds to comparative learning when the source class contains all functions from domain to label set, that is $\mathcal{S} = \mathcal{Y}^{\mathcal{X}}$ (Ben-David and Urner, 2014).

The study that introduced the comparative PAC learning framework provided a first upper and first lower bound for the sample complexity $\mathrm{n}_{\mathcal{S},\mathcal{B}}^{\mathrm{gen}}(\cdot, \cdot)$ of this learning problem in terms of the *mutual VC dimension* $\mathrm{vc}(\mathcal{S}, \mathcal{B})$ of the two classes involved. The mutual VC dimension $\mathrm{vc}(\mathcal{H}, \mathcal{H}')$ of two hypothesis classes $\mathcal{H}$ and $\mathcal{H}'$ is defined as the largest possible size of a set of points that is simultaneously shattered by both classes (see Appendix, Section A for a formal definition). The following citation summarizes these initial bounds, omitting log-factors.

**Theorem 2 (Hu and Peale (2023), Theorem 3.1)** *Let $\mathcal{S}, \mathcal{B} \subseteq \{0, 1, \star\}^{\mathcal{X}}$ be two hypothesis classes with $\mathrm{vc}(\mathcal{S}, \mathcal{B}) \geq 2$. Then the sample complexity of comparatively learning the pair $(\mathcal{S}, \mathcal{B})$ satisfies:*

$$\Omega\left(\frac{\mathrm{vc}(\mathcal{S}, \mathcal{B}) + \log(1/\delta)}{\epsilon}\right) = \mathrm{n}_{\mathcal{S},\mathcal{B}}^{\mathrm{gen}}(\epsilon, \delta) = \tilde{O}\left(\frac{\mathrm{vc}(\mathcal{S}, \mathcal{B}) + \log(1/\delta)}{\epsilon^2}\right).$$

We view these bounds as a starting point for our investigations. We note that while both bounds are in terms of the same combinatorial complexity parameter relating the source and benchmark class, namely their mutual VC dimension, there is a gap in the $\frac{1}{\epsilon}$-dependences, namely linear versus quadratic. We will identify general conditions for both cases and exhibit how they depend on more fine grained relatedness properties between the two classes at hand.

## 3. Proper Comparative Learning

In this section we consider general multiclass classification tasks with a potentially countably infinite label set $\mathcal{Y}$. Furthermore, unless otherwise stated, we consider general, partial hypothesis classes for source and benchmark classes so that they are a subset of $\tilde{\mathcal{Y}}^{\mathcal{X}}$.

The initial comparative learning bounds in Theorem 2 hold for any learner, in particular learners that are not proper for the benchmark class. However, the comparative learning framework is especially suitable for modeling learning settings where we are interested in outputting a classifier with specific properties, which is most suitably modeled with the benchmark-proper setting. Moreover, we note that practical methods are typically set up to minimize an empirical loss over a training data set, and are thus ERM (or approximate ERM) methods for a specific class of predictors. We start by showing that for such methods the sample complexity can be significantly higher than what is stated in Theorem 2. For this, we define a novel relatedness parameter between the source and benchmark class, and term it the *one-sided graph dimension*.

**Definition 3 (One-sided Mutual Graph Dimension)** *A set $U \subseteq \mathcal{X}$ is one-sided graph-shattered by $(\mathcal{S}, \mathcal{B})$ if there exists a function $s \in \mathcal{S}$, which is a total function when restricted to $U$, such that for all $V \subseteq U$, there exists $b \in \mathcal{B}$ such that*

$$b(x) = s(x) \text{ for all } x \in V, \quad \text{and} \quad b(x) \neq s(x) \text{ for all } x \in U \backslash V.$$

*The* one-sided mutual graph dimension *of $(\mathcal{S}, \mathcal{B})$ is the maximum size of a set that can be one-sided graph-shattered and is denoted by $d_G^{\rightarrow}(\mathcal{S}, \mathcal{B})$. We define $d_G^{\rightarrow}(\mathcal{S}, \mathcal{B}) = \infty$ if the pair $(\mathcal{S}, \mathcal{B})$ one-sided graph-shatters sets of arbitrary size.*

For two partial binary classes, it is easy to see that the one-sided mutual graph dimension is always an upper bound to their mutual VC dimension, i.e., $\mathrm{vc}(\mathcal{S}, \mathcal{B}) \leq d_G^{\rightarrow}(\mathcal{S}, \mathcal{B})$. The next observation shows that the gap between these two parameters can be arbitrarily large.

**Observation 1** *There exist total, binary classes $\mathcal{S}$ and $\mathcal{B}$ with $\mathrm{vc}(\mathcal{S}, \mathcal{B}) = 0$, while $d_G^{\rightarrow}(\mathcal{S}, \mathcal{B}) = \infty$.*

**Proof** Let $\mathcal{X}$ be an infinite set, e.g., $\mathcal{X} = \mathbb{N}$. Consider the classes $\mathcal{S} = \{h_0\}$, where $h_0(x) = 0$ for all $x \in \mathcal{X}$, and $\mathcal{B} = \{0, 1\}^{\mathcal{X}}$ (or any binary class with $\mathrm{vc}(\mathcal{B}) = \infty$). We get $\mathrm{vc}(\mathcal{S}, \mathcal{B}) = 0$. To see that $d_G^{\rightarrow}(\mathcal{S}, \mathcal{B}) = \infty$, consider any set $U \subseteq \mathcal{X}$ of any size that is shattered by $\mathcal{B}$. For any $V \subseteq U$, choose $s = h_0 \in \mathcal{S}$, and $b \in \mathcal{B}$ such that $b$ is 0 on $V$ and is 1 on $U \backslash V$. Such $b$ exists because $U$ is shattered by $\mathcal{B}$. Thus any such set of arbitrary size is shattered by $(\mathcal{S}, \mathcal{B})$ in the sense of Definition 3, and thus the one-sided mutual graph dimension is $\infty$. ∎

### Benchmark-ERM comparative learning

We start by analyzing the sample complexity of benchmark-ERM learners for comparative learning. We define the *benchmark-ERM sample complexity* to be the smallest sample size function $\mathrm{n}_{\mathcal{S}, \mathcal{B}}^{\mathrm{ERM}} : (0, 1)^2 \rightarrow \mathbb{N}$ such that the conditions of Definition 1 are satisfied for all ERM learners for the benchmark class with this function.

The following Theorem summarizes the upper and lower bounds for benchmark-ERM comparative learning and establishes that this setting is governed by the one-sided mutual graph dimension:

**Theorem 4** *For any pair of partial classes $(\mathcal{S}, \mathcal{B})$ the sample complexity of benchmark-ERM is determined by the one-sided graph dimension and satisfies:*

$$\Omega \left( \frac{d_G^{\rightarrow}(\mathcal{S}, \mathcal{B}) + \log(\frac{1}{\delta})}{\epsilon} \right) = \mathrm{n}_{\mathcal{S}, \mathcal{B}}^{\mathrm{ERM}}(\epsilon, \delta) = \tilde{\mathcal{O}} \left( \frac{d_G^{\rightarrow}(\mathcal{S}, \mathcal{B}) + \log(\frac{1}{\delta})}{\epsilon^2} \right).$$

As noted above, the one-sided mutual graph dimension always upper bounds the mutual VC dimension of two partial binary hypothesis classes and Observation 1 above shows that the gap between these two parameters can be arbitrarily large. This implies that, for the case of ERM learners, our lower bound is a significant strengthening of the lower bound for arbitrary learners from Theorem 2. It also implies that the sample complexity of ERM for comparative learning can differ significantly from the general sample complexity of the task (where no properness restrictions are imposed). The theorem is established through the bounds in Lemma 5 and 6 below.

Examples 2 and 3 in Appendix B illustrate how benchmark-ERM learning in the comparative setting relates to learning the benchmark class in the usual PAC setting.

We now proceed to prove Theorem 4:

**Lemma 5** *For any pair of partial classes $(\mathcal{S}, \mathcal{B})$, the sample complexity of benchmark-ERM satisfies*

$$\mathrm{n}_{\mathcal{S},\mathcal{B}}^{\mathrm{ERM}}(\epsilon, \delta) = \Omega\left(\frac{d_G^{\rightarrow}(\mathcal{S}, \mathcal{B}) + \log(\frac{1}{\delta})}{\epsilon}\right)$$

**Proof** [Proof sketch] We need to show that there exist benchmark-ERM learners that do not succeed in the sense of Definition 1 with less than the stated sample sizes. Let $d = d_G^{\rightarrow}(\mathcal{S}, \mathcal{B})$ and let $U = \{x_1, \ldots, x_d\}$ be a set that is one-sided graph-shattered by $(\mathcal{S}, \mathcal{B})$ and also let $s^* \in S$ be the total function on $U$ that witnesses this shattering. By definition, this means there exists $b^* \in B$ such that $s^*(x) = b^*(x)$ for all $x \in U$. We now consider distributions over $\mathcal{X} \times \mathcal{Y}$ whose marginal on $\mathcal{X}$ has support included in $U$ with labeling $s^*$. Since $s^* = b^*$ on $U$ we have $\inf_{b \in B} \Pr[b(x) \neq s^*(x)] = 0$ for all such distributions. Consider a "bad" benchmark-ERM learner that for sample points $V \subseteq U$, returns $b \in B$ with $b(x) = s^*(x)$ on $V$ and $b(x) \neq s^*(x)$ on $U \backslash V$. Such $b$ exists due to the definition of shattering. Also, note that this is a valid ERM since it has error 0 on samples. Now, for a given $\epsilon > 0$, we we define a distribution with marginal support in $U$ as follows:

$$\Pr[x_1] = 1 - 2\epsilon, \Pr[x_i] = \frac{2\epsilon}{d-1} \forall i \in [d] \backslash \{x_1\}$$

Standard arguments show that as long as the learner receives less than $\frac{d-1}{12\epsilon} + \frac{1}{4\epsilon}\log(\frac{1}{\delta})$ many samples from this distribution, it will not see a significant portion of the "light" points in the training sample, and thus misclassify them, yielding an error larger than $\epsilon$. The remaining details, which follow the same line of argument as the proof of Theorem 9 by Daniely et al. (2015), can be found in the full version in Appendix C. ∎

Next, we show the one-sided graph dimension also yields an upper bound on $\mathrm{n}_{\mathcal{S},\mathcal{B}}^{\mathrm{ERM}}$ (and thus $\mathrm{n}_{\mathcal{S},\mathcal{B}}^{\mathrm{prop}}$).

**Lemma 6** *For any pair of partial classes $(\mathcal{S}, \mathcal{B})$ the sample complexity of benchmark-ERM satisfies*

$$\mathrm{n}_{\mathcal{S},\mathcal{B}}^{\mathrm{ERM}}(\epsilon, \delta) = \tilde{\mathcal{O}}\left(\frac{d_G^{\rightarrow}(\mathcal{S}, \mathcal{B}) + \log(\frac{1}{\delta})}{\epsilon^2}\right).$$

**Proof** We will show that the benchmark class $\mathcal{B}$ enjoys uniform convergence (see Definition 21) when restricted to the set of distributions that are realizable by the source class $\mathcal{S}$. This implies that any benchmark-ERM learner is successful in the sense of Definition 1. To see this, let us view both source class $\mathcal{S}$ and benchmark class $\mathcal{B}$ as collections of subsets of $\mathcal{X} \times \mathcal{Y}$. Note that realizability with respect to the source class $\mathcal{S}$ implies that we are only considering distributions $P$ over $\mathcal{X} \times \mathcal{Y}$ whose support is included in one of the sets $s \in \mathcal{S}$, let's denote this set $s^*$. The benchmark-proper comparative learning task then is to choose a set $b \in \mathcal{B}$ with largest (in terms of probability weight) intersection with this set $s^*$.

Note that the definition of the one-sided graph dimension implies that for each $s \in \mathcal{S}$, the VC dimension of the collection of subsets $\mathcal{B}_s = \{s \cap b \mid b \in \mathcal{B}\}$ is finite. Moreover, the VC dimension of these collections is uniformly upper bounded by the one sided mutual graph dimension. That is $\mathrm{vc}(\mathcal{B}_s) \leq d_G^{\rightarrow}(\mathcal{S}, \mathcal{B})$ for all $s \in \mathcal{S}$. This implies that, restricted to the class of distributions realizable by $\mathcal{S}$, the benchmark class enjoys uniform convergence. This yields the upper bound on the sample complexity of any benchmark-ERM learner stated in the theorem. ∎

**General benchmark-proper comparative learning**

The lower bound in Theorem 4 is established through worst case benchmark-ERM learners, which are a special type of benchmark-proper learners. One might ask whether allowing general benchmark-proper learners yields a lower sample complexity. In Lemma 7 below we provide an example of two hypothesis classes for which the sample complexity lower bound in Theorem 4 holds *for any benchmark-proper learning algorithm*. As a corollary, we show that there exist classes (and in particular binary total classes) that are comparatively learnable, but cannot be learned by any benchmark-proper learner.

**Lemma 7** *For any $d \in \mathbb{N}$, there exists a pair $(\mathcal{S}, \mathcal{B})$ of total, binary classes with $d_G^{\rightarrow}(\mathcal{S}, \mathcal{B}) \geq \lfloor \frac{d}{2} \rfloor$ and $\mathrm{vc}(\mathcal{S}, \mathcal{B}) = 0$ such that $\mathrm{n}_{\mathcal{S},\mathcal{B}}^{\mathrm{prop}}(\epsilon, \delta) = \Omega \left( \dfrac{d_G^{\rightarrow}(\mathcal{S}, \mathcal{B}) + \log(\frac{1}{\delta})}{\epsilon} \right).$*

**Proof** *[Proof sketch] Let $\mathcal{X}_d = \{x_1, \ldots, x_d\}$, $\mathcal{X} = \{x_0\} \cup \mathcal{X}_d$. Let $\mathcal{S} = \{h_1\}$ consist of only the all-1 function and let $\mathcal{B} = \{1_{U \cup \{x_0\}} \mid U \subseteq \mathcal{X}_d, |U| = d/2\}$ where we have used $1_U$ to denote a function defined as $1_U(x) = \mathbb{1}\left[x \in U\right]$. It is not hard to see that $d_G^{\rightarrow}(\mathcal{S}, \mathcal{B}) = d/2$. Fix any benchmark-proper algorithm and let $m$ be its sample complexity. Fix $\epsilon > 0$ and $0 < \delta \leq \epsilon$. Let $\epsilon' = \frac{2\epsilon}{1-\exp(-1/24)}$. For $U \subseteq \mathcal{X}_d$ with $|U| = d/2$, let $P_U$ be a distribution on $\mathcal{X}$ such that $P_U(x_0) = 1 - 4\epsilon'$ and $P_U(x) = \frac{8\epsilon'}{d}$ for $x \in U$. Then we can show that for the learner to be successful on all of these distributions, we need $\Omega \left( \dfrac{d_G^{\rightarrow}(\mathcal{S}, \mathcal{B}) + \log(\frac{1}{\delta})}{\epsilon} \right)$ samples. The main intuition is that while the true labeling is constant 1, a benchmark proper learner that has only seen a quarter of the relevant domain points, would need to "guess" where the other quarter with positive probability mass sits, which is not possible. The technical details of this argument can be found in the full proof in Appendix C.* ∎

**Corollary 8** *There exists a pair $(\mathcal{S}, \mathcal{B})$ of total, binary classes that is comparatively learnable, but cannot be learned by any benchmark-proper learner.*

**Proof** Let $(\mathcal{S}_d, \mathcal{B}_d)$ be as in Lemma 7 defined on $\mathcal{X}_d$. Let $\mathcal{X} = \{x_0\} \bigcup \cup_{d \in \mathbb{N}} \mathcal{X}_d$ where $\mathcal{X}_d$'s are mutually exclusive. Let $\mathcal{B} = \cup_{d \in \mathbb{N}} \mathcal{B}_d$ where $b \in \mathcal{B}_d$ labels each point in $\mathcal{X}_{d'}$ with 1 for $d' \neq d$ and let $\mathcal{S} = \{h_1\}$. Then, for each $d \in \mathbb{N}$ there exists a distribution such that any proper learner needs $\Omega(\frac{d+\log(\frac{1}{\delta})}{\epsilon})$ samples. Considering the limit $d \to \infty$ establishes the impossibility for benchmark-proper learning. However, $\mathrm{vc}(\mathcal{S}, \mathcal{B}) = 0$ and thus, the pair is comparatively learnable. ∎

**Linear versus quadratic dependence on the error parameter**

Our upper and lower bound in Theorem 4 still exhibit a gap in terms of their dependence on the error parameter $\frac{1}{\epsilon}$ (linear versus quadratic). We now show that bothrates can occur for proper comparative learning and provide a general condition that enforces quadratic dependence (a slow rate). When the two classes coincide (that is, $\mathcal{S} = \mathcal{B}$), the comparative learning task reduces to proper PAC learning in the realizable case (see Definition 19 in the Appendix). Note that in case both the source and benchmark are total binary classes the one-sided mutual graph dimension corresponds to the VC dimension of the benchmark class, and thus in case $\mathcal{S} = \mathcal{B}$ the proper learning sample complexity is $\mathrm{n}_{\mathcal{S},\mathcal{B}}^{\mathrm{prop}}(\epsilon, \delta) = \tilde{\Theta}\left( (d_G^{\rightarrow}(\mathcal{S}, \mathcal{B}) + \log(\frac{1}{\delta}))/(\epsilon) \right)$, corresponding to the lower bound in Theorem 4. On

the other hand, if the source class contains all binary functions $\mathcal{S} = \{0, 1\}^{\mathcal{X}}$, then the comparative learning task corresponds to PAC learning with deterministic labels, and it has been shown that there are classes whose ERM sample complexity and proper learning sample complexity exhibit quadratic dependence on $\frac{1}{\epsilon}$ (Ben-David and Urner, 2014; Ben-David and Ben-David, 2011).

Interestingly, whether the proper learning sample complexity exhibits a $\frac{1}{\epsilon}$ or $\frac{1}{\epsilon^2}$ dependence hinges on the relatedness between the two classes rather than merely on the complexity of the source class (one might expect that a more complex source class corresponds to a more challenging learning problem).

It has been shown that proper learning can require quadratic dependence on $\frac{1}{\epsilon}$ even if the true labeling of the distribution is known to the learner (Ben-David and Ben-David, 2011). This corresponds to comparative learning with a source class that is a singleton.

With the following definition and theorem we provide a sufficient condition for quadratic dependence in benchmark-proper comparative learning setting. We show that they are induced by the following relatedness parameter, which we term the *mutual star dimension* of source and benchmark class. It is closely related to the hollow star number (Bousquet et al., 2020). For a hypothesis $h$ and a set $K$, let $h|_K$ be the restriction of h to $K$. Similarly, for a hypothesis class $\mathcal{H}$, let $\mathcal{H}|_K$ be a class including hypotheses from $\mathcal{H}$ restricted to $K$.

**Definition 9 (Mutual Star Dimension)** *A mutual star of size $k$ for a pair of classes $\mathcal{S}, \mathcal{B} \subseteq \tilde{\mathcal{Y}}^{\mathcal{X}}$ consists of a set of $k + 1$ points $K = \{x_1, \ldots, x_{k+1}\} \subseteq \mathcal{X}$, $s \in \mathcal{S}$ and $b_1, \ldots, b_{k+1} \in \mathcal{B}$ with the following properties:*

- *$s|_K$ is a total function,*

- *$s|_K$ is not realizable by $\mathcal{B}|_K$, and*

- *$s(x_i) \neq b_i(x_i)$ and $s(x_j) = b_i(x_j)$ for each $i, j \in [k + 1]$ with $j \neq i$.*

*We define the mutual star dimension of $(\mathcal{S}, \mathcal{B})$, denoted by $\mathfrak{s}(\mathcal{S}, \mathcal{B})$, to be size of the smallest mutual star of this pair, if at least one such star exists. Otherwise, the mutual star dimension is defined to be $\mathfrak{s}(\mathcal{S}, \mathcal{B}) = \infty$.*

**Theorem 10** *For any pair of hypothesis classes $(\mathcal{S}, \mathcal{B})$, with finite mutual star dimension $\mathfrak{s}(\mathcal{S}, \mathcal{B}) = k$, $\mathrm{n}^{\mathrm{prop}}_{\mathcal{S}, \mathcal{B}}(\epsilon, \delta) = \Omega(\frac{1}{k} \frac{1}{\epsilon^2} \log(\frac{1}{\delta}))$.*

**Proof** Fix $0 < \epsilon < \frac{1}{k+1}$ and $0 < \delta < \frac{1}{4}$. Consider a mutual star of size $k$ with parameters $K = \{x_1, \ldots, x_{k+1}\} \subseteq \mathcal{X}$, $s \in \mathcal{S}$ and $b_1, \ldots, b_{k+1} \in \mathcal{B}$ as defined in Definition 9. Consider a set of $k + 1$ distributions whose support lies in $K$ and are defined as follows: $P_i(x_i) = \frac{1+\epsilon}{k+1} - \epsilon$ and $P_i(x_j) = \frac{1+\epsilon}{k+1}$ for all $i, j \in [k + 1]$ with $j \neq i$. Then as $s$ is not realizable by the benchmark class, $OPT_i = \inf_{b \in \mathcal{B}} \mathcal{L}_{P_i}(b) = \frac{1+\epsilon}{k+1} - \epsilon$ which is attained by $b_i$. On the other hand, if the learner outputs any other hypothesis in $\mathcal{B}$ that differs with $s$ on any point other than $x_i$, it will occur loss of at least $\frac{1+\epsilon}{k+1}$ which is bigger than $OPT_i + \epsilon$ and means the learner is not a valid comparative learner. Thus, if the adversary picks the distribution uniformly random between $P_i$'s, a successful learner must find out which point $x_i$ has the smallest mass. We can thus reduce the *weighted dice problem* with $k$ sides to this learning problem, and the sample complexity of this problem has been shown to be lower bounded by $\Omega(\frac{1}{k} \frac{1}{\epsilon^2} \log(\frac{1}{\delta}))$ (Theorem 2 by Ben-David and Ben-David (2011)). ∎

The following simple example illustrates the phenomenon.

**Example 1** *Consider a domain containing two points $\mathcal{X} = \{x_0, x_1\}$, a source class $\mathcal{S} = \{h_{01}\}$, and benchmark class $\mathcal{B} = \{h_{00}, h_{11}\}$, where the index of the function indicates which labels the function assigns to the two points. The set $K = \mathcal{X}$, $s = h_{01}$ and $b_1 = h_{00}, b_2 = h_{11}$ form a mutual star of size 1 and thus $\mathfrak{s}(\mathcal{S}, \mathcal{B}) = 1$. Now consider the class of distributions that are realizable by the source (thus label $x_0$ with $0$ and $x_1$ with $1$) and assign probability weights $\frac{1}{2} \pm \epsilon$ to the two points, for all $\epsilon > 0$. A benchmark proper learner then has to estimate which of the two points has heavier probability weight, in order to choose the optimal classifier among $\{h_{00}, h_{11}\}$. The proper learning problem over this set of distributions thus reduces to estimating the bias of a coin flip, which is known to require sample sizes of $\Omega(1/\epsilon^2)$ (Anthony and Bartlett, 2002).*

## 4. General Comparative Learning

We now consider the general comparative learning setting, where no properness requirement is imposed. We start by again noting that both linear and quadratic dependence in the error parameter $\frac{1}{\epsilon}$ can occur even in the case of total binary classes: as noted above, the case where source and benchmark coincide, $\mathcal{S} = \mathcal{B}$, yields sample complexity $\mathrm{n}_{\mathcal{S},\mathcal{B}}^{\mathrm{gen}}(\epsilon, \delta) = \tilde{\Theta}(\frac{\mathrm{vc}(\mathcal{B}) + \log(1/\delta)}{\epsilon})$ in the realizable case (see Theorem 22 in Appendix A). The case where the source consists of all binary functions, $\mathcal{S} = \{0, 1\}^{\mathcal{X}}$ corresponds to agnostic PAC learning with deterministic labels. For binary total classes it has been shown that this setting has sample complexity with quadratic dependence $\Omega(1/\epsilon^2)$ if and only if the hypothesis class has infinite *diameter* (Ben-David and Urner, 2014). The diameter $\mathrm{diam}(\mathcal{H})$ of a total hypothesis class $\mathcal{H}$ is defined to be the largest set on which two functions from the class disagree:

$$\mathrm{diam}(\mathcal{H}) = \sup\{k \in \mathbb{N} \mid \exists U \subseteq \mathcal{X}, |U| = k, \exists h_1, h_2 \in \mathcal{H} \text{ such that } h_1(x) \neq h_2(x) \forall x \in U\}$$

Thus, for the benchmark class $\mathcal{B} = \{h_0, h_1\}$ containing only the constant $0$ and constant $1$ functions (or any benchmark class including $\{h_0, h_1\}$), the comparative learning sample complexity with source $\mathcal{S} = \{0, 1\}^{\mathcal{X}}$ (or any source class that shatters arbitrarily large sets) is $\mathrm{n}_{\mathcal{S},\mathcal{B}}^{\mathrm{gen}}(\epsilon, \delta) = \Omega(1/\epsilon^2)$.

We start by generalizing some of the results for agnostic learning under deterministic labels to partial classes to derive more fine grained bounds for comparative learning of partial classes.

### 4.1. Agnostic learning with deterministic labels for partial classes

We start by extending the notion of diameter to partial hypothesis classes. Recall that for a partial hypothesis $h \in \tilde{\mathcal{Y}}^{\mathcal{X}}$, we define the support of $h$ to be the subset of the domain, where $h$ assigns labels, that is $\mathrm{supp}(h) = \{x \in \mathcal{X} \mid h(x) \in \mathcal{Y}\}$.

**Definition 11 (Diameter and joint diameter for partial classes)** *Let $\mathcal{H} \subseteq \tilde{\mathcal{Y}}^{\mathcal{X}}$ be a partial binary hypothesis class. We define the* diameter *of the partial class as*

$$\mathrm{diam}(\mathcal{H}) = \sup_{h,h' \in \mathcal{H}} |\{x \in \mathrm{supp}(h) \mid h(x) \neq h'(x)\}|.$$

*We further define the* joint diameter *of the partial class as*

$$\mathrm{diam}'(\mathcal{H}) = \sup_{h,h' \in \mathcal{H}} |\{x \in \mathrm{supp}(h) \cap \mathrm{supp}(h') \mid h(x) \neq h'(x)\}|.$$

Note that for any binary partial class $\mathcal{H}$ we have $\text{vc}(\mathcal{H}) \leq \text{diam}'(\mathcal{H}) \leq \text{diam}(\mathcal{H})$ where VC-dimension of a partial class, denoted by $\text{vc}(\mathcal{H})$, is defined in Appendix A. In case $\mathcal{H}$ is a total class, the two latter notions of diameter coincide and equal the diameter for total classes as defined above.

We will now show that bounded diameter implies a fast rate, $\tilde{O}(\frac{1}{\epsilon})$ while infinite joint diameter implies a slow rate of $\Omega(\frac{1}{\epsilon^2})$

**Theorem 12** *Let $\mathcal{H} \subseteq \tilde{\mathcal{Y}}^{\mathcal{X}}$ be a partial hypothesis class with finite diameter $\text{diam}(\mathcal{H}) = d < \infty$. Then the sample complexity of agnostically PAC learning $\mathcal{H}$ under deterministic labels is upper bounded by*

$$\mathcal{O}(\frac{d}{\epsilon}[\log(\frac{d}{\epsilon}) + \log(\frac{1}{\delta})])$$

**Proof** Let $h_0 \in \mathcal{H}$ be an arbitrary function in the hypothesis class. Similar to the proof of Theorem 9 in Ben-David and Urner (2014), we consider the following learner $\mathcal{A}$: given a sample $S = ((x_1, y_1), (x_2, y_2), \ldots (x_n, y_n))$ the learner outputs a classifier $f = \mathcal{A}(S)$ with $f(x) = y_i$ in case $x = x_i$ for some $i \in [n]$ and $f(x) = h_0(x)$ otherwise. Note that this is not a proper learner for the class $\mathcal{H}$ (even if $\mathcal{H}$ is a total class).

Now consider any deterministic distribution $P$ over $\mathcal{X} \times \mathcal{Y}$ and let $h^* \in \mathcal{H}$ be the hypothesis that achieves the approximation error of the class, that is $\mathcal{L}_P(h^*) = \text{opt}_P(\mathcal{H})$. A sample of size $\mathcal{O}(\frac{d}{\epsilon}[\log(\frac{d}{\epsilon}) + \log(\frac{1}{\delta})])$ contains every point with mass at least $\frac{\epsilon}{d}$ with probability more than $1 - \delta$. To see this, note that the probability of a point with mass at least $\frac{\epsilon}{d}$ not appearing in a sample of size $m$ is upper bounded by $(1 - \frac{\epsilon}{d})^m \leq \exp(-m\frac{\epsilon}{d})$. There are at most $\frac{d}{\epsilon}$ such points, thus, probability that at least one of them does not appear in the sample is upper bounded by $\frac{d}{\epsilon}\exp(-m\frac{\epsilon}{d})$ by a union bound. Setting this to be less than $\delta$ gives the desired results. We now condition the rest of the proof on this event and show that the learner achieves error at most $\mathcal{L}_P(f) \leq \text{opt}_P(\mathcal{H}) + \epsilon$ (the failure probability is thus bounded by $\delta$ as required).

Since the distribution $P$ is deterministic, the classifier $f = \mathcal{A}(S)$ will not make any mistake on points from $S$. Further, due to the finite diameter, outside of the sample the classifier $f$ disagrees with $h^*$ on at most $d = \text{diam}(\mathcal{H})$ points from the support $\text{supp}(h^*)$ of the optimal classifier (since $f$ classifies according to the default function $h_0 \in \mathcal{H}$ on these points). As we assumed that the sample contained all points with probability mass at least $\epsilon/d$, the $d$ points on which $f$ and $h^*$ disagree have joint mass at most $d \cdot \epsilon/d = \epsilon$. This implies $\mathcal{L}_P(f) \leq \mathcal{L}_P(h^*) + \epsilon$, which completes the proof. ∎

We now show that infinite joint diameter implies a slow rate for learning under deterministic labels:

**Theorem 13** *Let $\mathcal{H} \subseteq \tilde{\mathcal{Y}}^{\mathcal{X}}$ be a partial hypothesis class with infinite joint diameter $\text{diam}'(\mathcal{H}) = \infty$. Then the sample complexity of agnostically PAC learning $\mathcal{H}$ under deterministic labels is lower bounded by $\Omega\left(\frac{1}{\epsilon^2}\right)$.*

**Proof** Since $\text{diam}'(\mathcal{H}) = \infty$, for any $\epsilon > 0$, there exist two hypotheses $h_0, h_1 \in \mathcal{H}$ and a set $U$ with cardinality $\Omega(\frac{1}{\epsilon^3})$ such that $U \subseteq \text{supp}(h_0) \cap \text{supp}(h_1)$ and $h_0(x) \neq h_1(x)$ for all $x \in U$. We now consider the set of all label-deterministic distributions $P$ over $\mathcal{X} \times \mathcal{Y}$ with marginal support included in $U$, that is $\text{supp}(P_{\mathcal{X}}) \subseteq U$. Learning the class $\mathcal{H}$ with respect to this set distributions requires at least as many samples as learning the smaller class $\{h_0, h_1\}$. Thus, by Lemma 3 and Remark 5 in Ben-David and Urner (2014), restated in Lemma 23 in the Appendix, and noting that restrictions of $h_0$ and $h_1$ to $U$ are total hypotheses, the sample complexity is lower bounded by $\Omega\left(\frac{1}{\epsilon^2}\right)$. ∎

### 4.2. Fast and slow rates in general comparative learning

We now use the results from the previous section to derive novel bounds for the comparative learning scenario. We state out main results in this section for the case where the source and benchmark are both partial binary classes, but in Remark 18, we discuss how they can be generalized for general label spaces. As in the original work on the comparative setting, we will employ the *agreement class* of source and benchmark.

**Definition 14 (Agreement Class, due to Hu and Peale (2023))** *For two (partial) hypotheses $s$ and $b$, we define their* agreement hypothesis *as follows:*

$$a_{s,b}(x) = \begin{cases} y & s(x) = b(x) = y \in \mathcal{Y} \\ \star & \textit{otherwise} \end{cases}$$

*Furthermore, for two (partial) hypothesis classes $\mathcal{S}$ and $\mathcal{B}$, their* agreement hypothesis class *is defined as $A_{\mathcal{S},\mathcal{B}} = \{a_{s,b} \mid s \in \mathcal{S}, b \in \mathcal{B}\}$.*

Note that the agreement class is a partial class (except for degenerate cases) even if both $\mathcal{S}$ and $\mathcal{B}$ are total classes. It has been shown that the VC dimension of the agreement class coincides with the mutual VC dimension of the two involved classes, $\mathrm{vc}(A_{\mathcal{S},\mathcal{B}}) = \mathrm{vc}(\mathcal{S}, \mathcal{B})$, and that any agnostic learner for the agreement class comparatively learns source and benchmark in the sense of Definition 1 (Hu and Peale, 2023, Lemma 8, p. 72:18). Combining this with our upper bound in Theorem 12, and known sample complexity bounds for learning partial classes in the realizable case, yields:

**Theorem 15** *Let $\mathcal{S}, \mathcal{B}$ be two partial binary classes, and $d = \min\{\mathrm{vc}(\mathcal{S}), \mathrm{diam}(\mathcal{B}), \mathrm{diam}(A_{\mathcal{S},\mathcal{B}})\}$. Then the sample complexity of comparatively learning the pair $(\mathcal{S}, \mathcal{B})$ is upper bounded by*

$$\mathrm{n}_{\mathcal{S},\mathcal{B}}^{\mathrm{gen}}(\epsilon, \delta) = \tilde{O}\left(\frac{d + \log 1/\delta}{\epsilon}\right).$$

**Proof** If the source has finite VC dimension, then the sample complexity of learning this partial class in the realizable case is upper bounded by $\tilde{O}(\frac{\mathrm{vc}(\mathcal{S}) + \log 1/\delta}{\epsilon})$ (Alon et al., 2021), and realizably learning the source class satisfies the success criterion of comparative learning. Similarly, if the agreement class or the benchmark class have finite diameter, we can learn these classes with sample complexity $\tilde{O}(\frac{\mathrm{diam}(A_{\mathcal{S},\mathcal{B}}) + \log 1/\delta}{\epsilon})$ and $\tilde{O}(\frac{\mathrm{diam}(\mathcal{B}) + \log 1/\delta}{\epsilon})$ respectively, and in those cases the resulting learner satisfies the success criterion for comparative learning. Thus, the sample complexity of comparatively learning $(\mathcal{S}, \mathcal{B})$ will be determined by the sample complexity of learner that achieves smallest sample complexity among these three. ∎

One might ask how the three dimensions that play a role in Theorem 15 relate to each other. In Example 4, in Appendix B, we show that each of them can be finite while the other two are infinite. Finally, we identify a general condition which yields sample complexity growing quadratically in $\frac{1}{\epsilon}$.

**Definition 16 (Mutual VC-Diameter)** *The mutual VC-Diameter of a pair of partial classes $(\mathcal{S}, \mathcal{B})$ is defined as follows:*

$$\mathrm{vcdiam}(\mathcal{S}, \mathcal{B}) = \sup\{n \in \mathbb{N} \mid \exists U \subseteq \mathcal{X}, |U| = n, \mathcal{S} \textit{ VC-shatters } U, \textit{ and } \mathrm{diam}'(\mathcal{B}\mid_U) = n\}$$

**Theorem 17** *If* $\mathrm{vcdiam}(S, B) = \infty$, *the comparative sample complexity is* $\mathrm{n}^{\mathrm{gen}}_{\mathcal{S},\mathcal{B}}(\epsilon, \delta) = \Omega(\frac{1}{\epsilon^2})$.

**Proof** This bound follows directly from the quadratic lower bound of Theorem 13. ∎

**Remark 18** *The result of Theorem 15 can be generalized to any label space if we replace* $\mathrm{vc}(\mathcal{S})$ *by a dimension that characterizes PAC learning of* $\mathcal{S}$ *in the realizable setting for that label space. Theorem 17 can be generalized to any label space if we replace VC-shattering in Definition 16 with the following: for all* $A \subseteq U$, *there exist* $s \in \mathcal{S}$ *such that* $s(x) = b_0(x)$ *for* $x \in A$ *and* $s(x) = b_1(x)$ *for* $x \in U \backslash A$ *where* $b_0, b_1 \in \mathcal{B}$ *and* $b_0(x) \neq b_1(x)$ *for* $x \in U$.

## 5. Conclusion

We have provided novel upper and lower bounds for the comparative learning for both the benchmark-proper and the general learning setting. Distinguishing between these two has shown that there is a more varied and nuanced landscape of learning rates than what was known before. We note that while we have shown that both linear and quadratic dependence on $\frac{1}{\epsilon}$ occur under general conditions in both proper and general comparative learning, and have provided some sufficient conditions for these, we currently lack a full characterization of the rates. We hope that our work will inspire further investigations into this framework.

## Acknowledgments

Ruth Urner is also a faculty affiliate member at Toronto's Vector Institute. This research was funded by an NSERC discovery grant.

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

## Appendix A. Background Definitions and Concepts

**Definition 19 (PAC Learner (Valiant, 1984; Shalev-Shwartz and Ben-David, 2014))** *We say that a learner $\mathcal{A}$ is an (agnostic) PAC learner for hypothesis class $\mathcal{H}$ if for every $\epsilon, \delta > 0$, there is a sample-size $n(\epsilon, \delta)$ such that, for all $n \geq n(\epsilon, \delta)$, and all distributions $P$ we have*

$$\Pr_{S \sim P^n} [\mathcal{L}_P(\mathcal{A}(S)) \leq \mathrm{opt}_P(\mathcal{H}) + \epsilon] \geq 1 - \delta.$$

*We call $\mathcal{A}$ a PAC learner for $\mathcal{H}$ in the realizable case if the above requirement holds restricted to all distributions $P$ with with $\mathrm{opt}_P(\mathcal{H}) = 0$. If such a learner exists, the class $\mathcal{H}$ is called PAC learnable. The learner is called a proper PAC learner if it always outputs a function from $\mathcal{H}$ (and the class $\mathcal{H}$ is then called properly PAC learnable).*

It is well known that for total binary hypothesis classes, PAC learnability is equivalent to finiteness of the VC dimension of the class (Vapnik and Chervonenkis, 1971; Shalev-Shwartz and Ben-David, 2014).

**Definition 20 (VC dimension)** *A set $U = \{u_1, \ldots, u_d\} \subseteq \mathcal{X}$ is shattered by a total binary hypothesis class if $\mathcal{H}|_U = \{0, 1\}^{|U|}$, where $\mathcal{H}|_U = \{(h(u_1), \ldots, h(u_d)) : h \in \mathcal{H}\}$. The VC dimension of $\mathcal{H}$, denoted by $\mathrm{vc}(\mathcal{H})$, is defined as the maximum cardinality of a set that is shattered by $\mathcal{H}$. If $\mathcal{H}$ shatters sets of arbitrary size, $\mathrm{vc}(\mathcal{H}) = \infty$.*

Before stating the main results of PAC learning for binary hypothesis classes, we should state the definition of uniform convergence which plays an important role. It is easy to see any hypothesis class that satisfies the uniform convergence property is PAC learnable by any ERM (see, for example, Section 4.1 in Shalev-Shwartz and Ben-David (2014))

**Definition 21 (Uniform Convergence, Shalev-Shwartz and Ben-David (2014))** *We say a hypothesis class $\mathcal{H}$ satisfies the uniform convergence property, if there exists a function $n^{UC} : [0, 1]^2 \to \mathbb{N}$ such that for all $\epsilon, \delta > 0$ and for all distributions $P$, samples $S \sim P^n$ with $n \geq n^{UC}(\epsilon, \delta)$ satisfy $|\mathcal{L}_P(h) - \mathcal{L}_S(h)| < \epsilon$ for all $h \in \mathcal{H}$ simultaneously, with probability more than $1 - \delta$ over samples.*

**Theorem 22 (Theorem 6.7, 6.8 in Shalev-Shwartz and Ben-David (2014)) The fundamental theorem of binary classification**
*For any binary total hypothesis class $\mathcal{H}$ the followings are equivalent.*

- $\mathrm{vc}(\mathcal{H}) < \infty$.

- *$\mathcal{H}$ satisfies the uniform convergence property.*

- *$\mathcal{H}$ is agnostically PAC learnable with sample complexity $\Theta(\frac{\mathrm{vc}(\mathcal{H}) + \log(\frac{1}{\delta})}{\epsilon^2})$.*

- *$\mathcal{H}$ is realizably PAC learnable with sample complexity $\tilde{\Theta}(\frac{\mathrm{vc}(\mathcal{H}) + \log(\frac{1}{\delta})}{\epsilon})$*

- *$\mathcal{H}$ any ERM is a PAC learner (both in the agnostic and in the realizable setting) with sample the same sample complexity up to logarithmic factors.*

Alon et al. (2021) extended the definition of VC dimension for partial hypothesis classes. The definition is the same as above with the modification that a set $U$ is shattered if $\mathcal{H}|_U \supseteq \{0,1\}^{|U|}$, meaning that $\mathcal{H}$ can produce all binary patterns on $U$. They show that VC dimension still characterizes PAC learnability both in the agnostic and in the realizable setting. They show the sample complexity of agnostic learning is $\tilde{\Theta}(\frac{\text{vc}(\mathcal{H})+\log(\frac{1}{\delta})}{\epsilon^2})$ and the sample complexity of realizable learning is $\tilde{\Theta}(\frac{\text{vc}(\mathcal{H})+\log(\frac{1}{\delta})}{\epsilon})$ (Please refer to Appendix C in Alon et al. (2021) for more details). It is worth noting that they achieve their results using an improper algorithm and show there are learnable partial classes that cannot be learned by a proper learner. This is in contrast to the results for total classes where any learnable class can be learned by any ERM.

Ben-David and Urner (2014) studied the agnostic learning of binary classes when distributions have deterministic labels. Here we restate the following lemma from Ben-David and Urner (2014), which we use in proof of Theorem 13.

**Lemma 23 (Essentially Lemma 3 in Ben-David and Urner (2014))** *Let $0 < \epsilon < \frac{1}{4}$ and $0 < \delta < \frac{1}{32}$. Let $\mathcal{H}$ be a hypothesis class defined on $\mathcal{X}$ with $|\mathcal{X}| \geq \frac{1}{\epsilon^3}$ such that there exists two hypotheses $h_0$ and $h_1$ in $\mathcal{H}$ that disagree on $\mathcal{X}$. Then $(\epsilon/2, \delta)$-PAC learning this class in the agnostic setting with deterministic labels requires $\Omega(\frac{1}{\epsilon^2})$ samples.*

Note that Lemma 23 was originally proved for hypothesis classes that contain the constant all 1 and all 0 functions. However, as the authors in Ben-David and Urner (2014), the proof can be generalized to classes that contain two hypotheses that disagree everywhere. This can be proven by simply adapting the current proof to set of distributions that label $(1/2 + \epsilon)$-fraction of domain according to $h_0$ and the rest according to $h_1$ or vice versa.

The VC dimension and the above results on sample complexity of PAC learning have been generalized to also work for multiclass classification. Natarajan (1989) introduced two dimension referred to as the Natarjan dimension (Definition 24) and the Graph dimension (Definition 25), and showed they give lower bound and upper bound on sample complexity of multiclass classification, respectively. Ben-David et al. (1995) later showed that when label space is finite, the Natarjan dimension also gives an upper bound. Daniely et al. (2015) showed that the ERM sample complexity of multiclass learning is characterized by the graph dimension.

**Definition 24 (Natarjan dimension Natarajan (1989))** *A set $U \subseteq \mathcal{X}$ is Natarjan-shattered by a class $\mathcal{H} \subseteq \mathcal{Y}^{\mathcal{X}}$ if for any $A \subseteq U$, there exist two hypotheses $f$ and $g$ in $\mathcal{H}$ such $f(x) = g(x)$ for all $x \in A$ and $f(x) \neq g(x)$ for all $x \in U \backslash A$. The Natarjan dimension of a class $\mathcal{H}$ is the size of the largest set that is Natarjan-shattered by $\mathcal{H}$ and is denoted by $d_N(\mathcal{H})$. We say $d_N(\mathcal{H}) = \infty$ if $\mathcal{H}$ Natarjan-shatters sets with arbitrarily large cardinality.*

**Definition 25 (Graph dimension Natarajan (1989))** *A set $U \subseteq \mathcal{X}$ is Graph-shattered by a class $\mathcal{H} \subseteq \mathcal{Y}^{\mathcal{X}}$ if there exists a hypothesis $g \in \mathcal{Y}^{\mathcal{X}}$ such that for any $A \subseteq U$, there exist $h \in \mathcal{H}$ such that $h(x) = g(x)$ for all $x \in A$ and $h(x) \neq g(x)$ for all $x \in U \backslash A$. The Graph dimension of a class $\mathcal{H}$ is defined as the size of the largest set that is Graph-shattered by $\mathcal{H}$ and is denoted by $d_G(\mathcal{H})$. If $\mathcal{H}$ Graph-shatters sets with arbitrary size, then we defined $d_G(\mathcal{H}) = \infty$.*

We conclude this section by defining the mutual VC dimension of two hypothesis classes defined by Hu and Peale (2023).

**Definition 26 (Mutual VC Dimension)** *A set $U \subseteq \mathcal{X}$ is said to be mutually shattered by $(\mathcal{H}, \mathcal{H}')$ if it is VC-shattered by both $\mathcal{H}$ and $\mathcal{H}'$. The mutual VC dimension of $(\mathcal{H}, \mathcal{H}')$, denoted by $\mathrm{vc}(\mathcal{H}, \mathcal{H}')$, is the maximum cardinality of a set that is mutually shattered by $(\mathcal{H}, \mathcal{H}')$. If $(\mathcal{H}, \mathcal{H}')$ mutually shatter sets of arbitrary size, $\mathrm{vc}(\mathcal{H}, \mathcal{H}') = \infty$.*

## Appendix B. Examples

**Example 2** *When both source and benchmark are total, binary classes, $\mathcal{S}, \mathcal{B} \subseteq \{0,1\}^{\mathcal{X}}$, we have $d_G^{\rightarrow}(\mathcal{S}, \mathcal{B}) = \mathrm{vc}(\mathcal{B})$. Thus, by Theorem 4, any pair $(\mathcal{S}, \mathcal{B})$ of total binary classes is benchmark-ERM comparative learnable if and only if $\mathcal{B}$ is agnostic PAC learnable.*

**Example 3** *If $\mathcal{S} = \{h_0\}$ (the source contains only the all-0 classifier) and $\mathcal{B} = \{1,2\}^{\mathcal{X}}$, then $d_G^{\rightarrow}(\mathcal{S}, \mathcal{B}) = 0$. However, the Natarjan dimension and the usual Graph dimension of the benchmark are both infinite, i.e., $d_N(\mathcal{B}) = \infty$ and $d_G(\mathcal{B}) = \infty$. Please refer to Definition 24 and Definition 25 for definitions of these two dimensions. This, again by Theorem 4, implies that as soon as we allow more than two labels, benchmark-ERM comparative learning can be easier than agnostic PAC learning of the benchmark.*

**Example 4** *For $y \in \{0, 1, \star\}$, let $h_y$ denote a function that labels every point in the domain with $y$.*

- *Let $\mathcal{S} = \{h_0\}$ and $\mathcal{B} = \{h_0, h_1\}$ so that $A_{\mathcal{S},\mathcal{B}} = \{h_0, h_\star\}$. Then $\mathrm{vc}(\mathcal{S}) = 0 < \infty$, but $\mathrm{diam}(\mathcal{B}), \mathrm{diam}(A_{\mathcal{S},\mathcal{B}}) = \infty$.*

- *Let $\mathcal{S} = \{0,1\}^{\mathcal{X}}$ and $\mathcal{B} = \{h_0\}$ so that $A_{\mathcal{S},\mathcal{B}} = \{0,\star\}^{\mathcal{X}}$. Then $\mathrm{diam}(\mathcal{B}) = 0 < \infty$, but $\mathrm{vc}(\mathcal{S}), \mathrm{diam}(A_{\mathcal{S},\mathcal{B}}) = \infty$.*

- *Let $\mathcal{X} = \mathcal{X}_1 \cup \mathcal{X}_2$ where $\mathcal{X}_1$ and $\mathcal{X}_2$ are disjoint and both have an infinite cardinality. Let $\mathcal{S}$ be such that $\mathcal{S}|_{\mathcal{X}_1} = \{0,1\}^{\mathcal{X}_1}$ and $\mathcal{S}|_{\mathcal{X}_2} = \{h_1\}$. Moreover, let $\mathcal{B}$ be such that $\mathcal{B}|_{\mathcal{X}_1} = \{h_\star\}$ and $\mathcal{B}|_{\mathcal{X}_2} = \{0,\star\}^{\mathcal{X}_2}$. In this case $A_{\mathcal{S},\mathcal{B}} = \{h_\star\}$ and thus $\mathrm{diam}(A_{\mathcal{S},\mathcal{B}}) = 0 < \infty$. However, $\mathrm{vc}(\mathcal{S}), \mathrm{diam}(\mathcal{B}) = \infty$.*

## Appendix C. Full Proofs

**Proof** [Proof of Lemma 5] We need to show that there exist benchmark-ERM learners that don't succeed in the sense of Definition 1 with less than the stated sample sizes. Let $d = d_G^{\rightarrow}(\mathcal{S}, \mathcal{B})$ and let $U = \{x_1, \ldots, x_d\}$ be a set that is one-sided graph-shattered by $(\mathcal{S}, \mathcal{B})$ and also let $s^* \in S$ be the total function that witnesses this shattering. By definition, this means there exists $b^* \in B$ such that $s^*(x) = b^*(x)$ for all $x \in U$. Let the source function to be equal to $s^*$ on $U$. Since $s^* = b^*$ on $U$ this also means $\inf_{b \in B} \Pr[b(x) \neq s^*(x)] = 0$ for all distributions whose support is restricted to $U$. Consider a "bad" benchmark-ERM learner that for sample points $V \subseteq U$, returns $b \in B$ such that $b(x) = s^*(x)$ on $V$ and $b(x) \neq s^*(x)$ on $U \backslash V$. Such $b$ exists due to the definition of shattering. Also, note that this is a valid ERM since it has error 0 on samples. Define a distribution on $U$ as follows:

$$\Pr[x_1] = 1 - 2\epsilon, \Pr[x_i] = \frac{2\epsilon}{d-1} \forall i \in [d] \backslash \{x_1\}$$

We now prove the lower bound by showing that the bad benchmark-ERM needs to see as many samples as the maximum of $\frac{d-1}{6\epsilon}$ and $\frac{1}{4\epsilon} \log(\frac{1}{\delta})$ which is greater than their average, $\frac{d-1}{12\epsilon} + \frac{1}{4\epsilon} \log(\frac{1}{\delta})$.

Let $m$ be the size of the sample and let $\epsilon < \frac{1}{12}$ and $\delta < \frac{1}{100}$. We first show $m > \frac{d-1}{6\epsilon}$. To see this, suppose $m \leq \frac{d-1}{6\epsilon}$. Then using a simple Chernoff's bound, it is not hard to see that the sample points will be equal to $x_1$ except for at most $\frac{d-1}{2}$ of them with probability more than $\frac{1}{100} > \delta$. However, if this happens, since the learner has not seen at least $d - 1 - \frac{d-1}{2} = \frac{d-1}{2}$ light points, its probability of error will be at least $\frac{d-1}{2}\frac{2\epsilon}{d-1} = \epsilon$. This means the probability of failure is more than $\delta$ and thus the algorithm cannot $(\epsilon, \delta)$-PAC learn. To see that $m > \frac{1}{4\epsilon}\log(\frac{1}{\delta})$, note that with probability $(1 - 2\epsilon)^m$, the sample will contain only $x_1$. This means the learner will make an error on all light point which have mass $2\epsilon$, which is a failure. Thus, we need to ensure this probability is less than $\delta$. However, if $m \leq \frac{1}{4\epsilon}\log(\frac{1}{\delta})$, then $(1 - 2\epsilon)^m \geq e^{-4\epsilon m} \geq \delta$. ∎

**Proof** [Proof of Lemma 7] Here we assume $d$ is even and for odd $d$ we can replace $d$ with $d - 1$ in the rest of the proof. Let $\mathcal{X}_d = \{x_1, \dots, x_d\}$, $\mathcal{X} = \{x_0\} \cup \mathcal{X}_d$. Let $\mathcal{S} = \{h_1\}$ consists of only the all 1 function and let $\mathcal{B} = \{1_{U \cup \{x_0\}} : U \subseteq \mathcal{X}_d, |U| = d/2\}$ consists of all functions that label half the domain and $x_0$ with 1 and the rest with 0. Here we have used $1_U$ to denote a function defined as $1_U(x) = \mathbb{1}[x \in U]$. It is not hard to see that $d_G^{\rightarrow}(\mathcal{S}, \mathcal{B}) = d/2$. Fix any benchmark-proper algorithm and let $m$ be its sample complexity. Fix $\epsilon > 0$ and $0 < \delta \leq \epsilon$. Let $\epsilon' = \frac{2\epsilon}{1 - \exp(-1/24)}$. For $U \subseteq \mathcal{X}_d$ with $|U| = d/2$, let $P_U$ be a distribution on $\mathcal{X}$ such that $P_U(x_0) = 1 - 4\epsilon'$ and $P_U(x) = \frac{8\epsilon'}{d}$ for $x \in U$. First note that with probability $(1 - 4\epsilon')^m$ all samples equal $x_0$. In this case, for any learner that outputs a hypothesis corresponding $U$ then there exists a distribution $P_{\mathcal{X} \setminus U}$ which makes the error of learner 1. Thus, we need to make sure the probability of this event is less than $\delta$. Thus, we must have $e^{-8m\epsilon'} \leq (1 - 4\epsilon')^m < \delta$ which means $m = \Omega(\frac{\log(\frac{1}{\delta})}{\epsilon'}) = \Omega(\frac{\log(\frac{1}{\delta})}{\epsilon})$. We now show $m \geq \frac{d}{32\epsilon'} = \Omega(\frac{d}{\epsilon})$. Assume $m < \frac{d}{32\epsilon'}$.

Define $\mathcal{P}_k(A) := \{E \subseteq A : |E| = k\}$. For a sequence $S_\mathcal{X}$, let $\text{set}(S_\mathcal{X})$ denote its unique elements. We choose $U \in \mathcal{P}_{d/2}(\mathcal{X}_d)$ uniformly and get $m$ samples from $P_U$. We would like to lower bound the following:

$$\mathbb{E}_{U \sim \text{Unif}(\mathcal{P}_{d/2}(\mathcal{X}_d))} \mathbb{E}_{S_\mathcal{X} \sim P_U^m}[\mathcal{L}_{P_U}(\mathcal{A}(S_\mathcal{X}))] \tag{1}$$

We can decompose $U$ in the above into two parts. The part that is present in our sample and the rest of it. The number of sample points that fall into $\mathcal{X}_d$ are distributed as $N \sim \text{Bin}(m, 4\epsilon')$. Then the samples are distributed as $S_\mathcal{X} \sim \text{Unif}(\mathcal{X}_d^N)$ where we have ignored the samples that are $x_0$ for simplicity. The rest of the support is chosen uniformly from rest of the domain. However, note that the sample is a sequence and not a set. Thus, only $l = |\text{set}(S_\mathcal{X})| \leq N$ points have been chosen so far. Thus, the rest of the support is chosen as $A \sim \text{Unif}(\mathcal{P}_{d/2-l}(\mathcal{X}_d \setminus \text{set}(S_\mathcal{X})))$. Therefore, we can equivalently write 1 in the following form.

$$\mathbb{E}_N \mathbb{E}_{S_\mathcal{X}} \mathbb{E}_A[\mathcal{L}_{P_{S_\mathcal{X} \cup A}}(\mathcal{A}(S_\mathcal{X}))] \tag{2}$$

$$\geq \mathbb{E}_N \mathbb{E}_{S_\mathcal{X}} \mathbb{E}_A[\mathcal{L}_{P_{S_\mathcal{X} \cup A}}(\mathcal{A}(S_\mathcal{X})) | N \leq d/4] \Pr[N \leq d/4] \tag{3}$$

Using a Chernoff bound, we can see that $\Pr[N \geq d/4] \leq \exp(-\frac{d}{24}) \leq \exp(-\frac{1}{24})$ for $m \leq \frac{d}{32\epsilon'}$. Fix any $N \leq d/4$ and $S_\mathcal{X} \in \mathcal{X}_d^N$. Note that $|\text{set}(S_\mathcal{X})| \leq N \leq d/4$. Let $1_{\{x_0\} \cup V} = \mathcal{A}(S_\mathcal{X})$. W.l.o.g. we assume $V = \text{set}(S_\mathcal{X}) \cup B$ because the learner makes a mistake on any point in the sample that is not in its output set. Let $r = d/2 - |\text{set}(S_\mathcal{X})|$ be the size of the rest of the support which satisfies $r \geq d/4$. Then $\mathbb{E}_A|A \cap B| = \mathbb{E}\sum_{b \in B} 1_{b \in A} = r \Pr[b \in A] = \frac{r}{2}$ and $\mathbb{E}_A[A \setminus B] = \frac{r}{2}$. Therefore, $\mathbb{E}_A[\mathcal{L}_{P_{S_\mathcal{X} \cup A}}(\mathcal{A}(S_\mathcal{X}))] = \frac{r}{2}\frac{4\epsilon'}{d/2} \geq \epsilon'$. Putting it all together we have:

$$\mathbb{E}_{U \sim \text{Unif}(\mathcal{P}_{d/2}(\mathcal{X}_d))} \mathbb{E}_{S_\mathcal{X} \sim P_U^m}[\mathcal{L}_{P_U}(\mathcal{A}(S_\mathcal{X}))] \geq (1 - e^{-\frac{1}{24}})\epsilon' = 2\epsilon$$

Therefore, there exists $U$ such that $\mathbb{E}_{S_{\mathcal{X}} \sim P_U^m}[\mathcal{L}_{P_U}(\mathcal{A}(S_{\mathcal{X}}))] \geq 2\epsilon$. Thus, $\Pr_{S_{\mathcal{X}} \sim P_U^m}[\mathcal{L}_{P_U}(\mathcal{A}(S_{\mathcal{X}})) > \epsilon] \geq \epsilon \geq \delta$, which contradicts $(\epsilon, \delta)$-comparative learning. Thus, $m = \Omega(\frac{d + \log(\frac{1}{\delta})}{\epsilon})$. ∎

## Appendix D. Additional ERM Bounds in Terms of the Mutual Graph Dimension

In this section we define a new dimension which gives us upper bounds for the sample complexity of both source-ERM and benchmark-ERM. To do this, we first define an agreement loss class. Note that this is different from what is defined as the agreement class by Hu and Peale (2023).

**Definition 27 (Agreement Loss Class)** *For classes $\mathcal{S}, \mathcal{B} \subseteq \tilde{\mathcal{Y}}^{\mathcal{X}}$ and for any $s \in \mathcal{S}$ and $b \in \mathcal{B}$, define the agreement function $a_{s,b} : \mathcal{X} \times \tilde{\mathcal{Y}} \to \{0, 1\}$ as follows:*

$$a_{s,b}(x, y) = \begin{cases} 0 & s(x) = b(x) = y \in \mathcal{Y} \\ 1 & otherwise \end{cases} \tag{4}$$

*Also let $A_{\mathcal{S}, \mathcal{B}} = \{a_{s,b} : s \in \mathcal{S}, b \in \mathcal{B}\}$ denote the agreement class.*

Next we define the mutual graph dimension of $(\mathcal{S}, \mathcal{B})$. As we will show this dimension controls VC dimension of the agreement loss class.

**Definition 28 (Mutual Graph Dimension)** *A set $U \subseteq \mathcal{X}$ is mutually G-shattered by $(\mathcal{S}, \mathcal{B})$ if there exists a total function $g : U \to \mathcal{Y}$ such that for all $V \subseteq U$, there exists $s \in \mathcal{S}$ and $b \in \mathcal{B}$ such that:*

$$\forall x \in V : s(x) = b(x) = g(x)$$

*and*

$$\forall x \in U \backslash V : s(x) \neq g(x) \vee b(x) \neq g(x)$$

*The mutual graph dimension of $(\mathcal{S}, \mathcal{B})$, denoted by $d_G(S, B)$, is defined as the maximum size of a set that can be mutually G-shattered.*

**Lemma 29** *For any two hypothesis classes $(\mathcal{S}, \mathcal{B})$, $d_G(\mathcal{S}, \mathcal{B}) = \text{vc}(A_{\mathcal{S}, \mathcal{B}})$*
**Proof** *The proof is straightforward if you note that:*

$$s(x) = b(x) = g(x) \text{ if and only if } a_{s,b}(x, g(x)) = 0,$$

*and equivalently,*

$$s(x) \neq g(x) \vee b(x) \neq g(x) \text{ if and only if } a_{s,b}(x, g(x)) = 1$$

*Define $(U, g(U)) := \{(x, g(x)) : x \in U\}$. Then a set $U$ is mutually G-shattered by $(\mathcal{S}, \mathcal{B})$ with $g$ as a witness iff $(U, g(U))$ is $VC$-shattered by $A_{\mathcal{S}, \mathcal{B}}$.* ∎

Now we are ready to see how we can convert a proper learner for agnostic PAC learning of the agreement loss class to a benchmark-proper or source-proper comparative learner.

**Theorem 30** *A pair $(\mathcal{S}, \mathcal{B})$ can be both benchmark-properly and source-properly learned in the comparative setting if $d_G(\mathcal{S}, \mathcal{B}) = \mathrm{vc}(A_{\mathcal{S},\mathcal{B}}) < \infty$. Furthermore, the sample complexity of learners achieving this task is $\mathcal{O}(\dfrac{d_G(\mathcal{S}, \mathcal{B}) + \log(\frac{1}{\delta})}{\epsilon^2})$.*

**Proof** *Fix any $\epsilon, \delta > 0$, and any distribution $P$ on $\mathcal{X} \times \mathcal{Y}$ realizable with respect to $\mathcal{S}$. First, we can see that for any $a_{s,b}$ we have:*

$$L(a_{s,b}) := \Pr[a_{s,b}(x, y) \neq 0] = \Pr[s(x) \neq y \vee b(x) \neq y] \tag{5}$$

*Note that there does not necessarily exist a hypothesis $a_{s,b}$ with $L(a_{s,b}) = 0$. However, since $\mathrm{vc}(A_{\mathcal{S},\mathcal{B}}) < \infty$, by Theorem 22, we can learn $A_{\mathcal{S},\mathcal{B}}$ with $m = \mathcal{O}(\frac{\mathrm{vc}(A_{\mathcal{S},\mathcal{B}}) + \log(\frac{1}{\delta})}{\epsilon^2})$ by a **proper** algorithm $\mathcal{A}$. Let $S^m \in ((\mathcal{X} \times \mathcal{Y}) \times \{0\})^m$. Let $a_{s',b'} = \mathcal{A}(S^m)$.*

$$\begin{aligned}
\Pr[b'(x) \neq y] &\leq \Pr[s'(x) \neq y \vee b'(x) \neq y] \\
&= \Pr[a_{s',b'}(x, y) \neq 0] \\
&\leq \inf_{a_{s,b}} \Pr[a_{s,b}(x, y) \neq 0] + \epsilon \\
&= \inf_{s \in \mathcal{S}, b \in \mathcal{B}} \Pr[s(x) \neq y \vee b(x) \neq y] + \epsilon \\
&= \inf_{b \in \mathcal{B}} \Pr[b(x) \neq y] + \epsilon
\end{aligned}$$

*The inequality is due to the PAC guarantees of $\mathcal{A}$ and the last equality follows from the fact that $P$ is realizable with respect to $\mathcal{S}$. The same calculations are true if we replace $b'$ with $s'$ in the LHS.* ∎

**Corollary 31** *A pair $(\mathcal{S}, \mathcal{B})$ can be comparatively PAC learned by any source-ERM or benchmark-ERM if $d_G(\mathcal{S}, \mathcal{B}) = \mathrm{vc}(A_{\mathcal{S},\mathcal{B}}) < \infty$. Furthermore, the sample complexity of those learners is $\mathcal{O}(\dfrac{d_G(\mathcal{S}, \mathcal{B}) + \log(\frac{1}{\delta})}{\epsilon^2})$.*

**Proof** *Note that $a_{s',b'} \in \arg\min_{a_{s,b}} \frac{1}{n} \sum_{i=1}^{n} \mathbb{1}[a_{s,b}(x, y) \neq 0]$ if $b' \in \arg\min_{b \in \mathcal{B}} \frac{1}{n} \sum_{i=1}^{n} \mathbb{1}[b(x_i) \neq y_i]$ and $s' \in \arg\min_{s \in \mathcal{S}} \frac{1}{n} \sum_{i=1}^{n} \mathbb{1}[s(x_i) \neq y_i]$. Thus, if $b'(s')$ is an empirical risk minimizer among $\mathcal{B}(\mathcal{S}, respectively)$, then $a_{s',b'}$ is an empirical risk minimizer among $A_{\mathcal{S},\mathcal{B}}$ and thus is a proper learner by Theorem 22. The results follow by the arguments in proof of Theorem 30.* ∎

Although the bound for benchmark-ERM is not tight as we already proved a similar upper bound with the one-sided mutual graph dimension which is smaller than the mutual graph dimension, the upper bound for source-ERM is new as such result is lacking.

