# OpenReview forum: "Refining the Sample Complexity of  Comparative Learning"
_algorithmiclearningtheory.org/ALT/2025/Conference — ALT 2025_

### Official Review · Reviewer_y9ZF · 2024-11-05
**Interesting paper but there are some concerns**

**Rating:** 7
**Confidence:** 4

**Review:**

This paper takes on an analysis of "comparative learning" which is an extension of PAC learning proposed recently by Hu and Peale (2023). The setting of comparative learning (as explained by Hu and Peale, 2023) considers a pair of hypotheses classes, a source class $\mathcal{S}$ and a benchmark class $\mathcal{B}$; so that realisable learning can be thought of as the case where $\mathcal{S}$ constrains the potential hypotheses that the ground-truth labeling is generated from, and agnostic learning can be thought of as the case where $\mathcal{B}$ constrains the set of hypotheses that the model output by the learner will be compared against. While classical PAC learning considers a single hypothesis class, Hu and Peale (2023) argue that separating these two roles of the hypothesis class, by postulating two separate classes as described, can lead to new ways to study and characterise the sample complexity in more learning tasks than can be studied via the classical (single hypothesis class) PAC learning setting.

As far as I can see, this paper under review proposes to further the work of Hu and Peale (2023), particularly the sample complexity analysis. The paper presents upper and lower bounds on the sample complexity of comparative learning, for each of two setting called respectively 'proper' and 'improper' learning. The said bounds depend on error and confidence parameters, and some quantity depending mutually on classes $\mathcal{S}$ and $\mathcal{B}$. The work of Hu and Peale (2023) proposed the mutual VC dimension, and this paper under review proposes some new quantities (mutual graph dimension, mutual star dimension, and so on).

I have assessed the paper against the following criteria, with the following outcomes:

- **Quality/clarity/readability:** The paper is relatively well-written. Some minor typos (see my detailed comments below). However, I have no big concerns with respect to these criteria. I am confident to say that the paper is readable for the most part.

- **Originality and significance:**  It would appear that this work is re-packaging and perhaps a small increment from that of Hu and Peale (2023). I am open to stand corrected on this, please let me know if I missed something. However, at least this indicates that there is a need to articulate in a clear way the novelties and/or improvements with respect to Hu and Peale (2023). For instance, if what this work calls "proper" comparative learning  (where the learner is required to output a predictor from the benchmark class) is a novel setting here, which was not considered by Hu and Peale (2023), this needs to be highlighted prominently and so does the associated analysis.
Also I'd like to ask the author(s) to address this question: Is not the gap between your upper and lower bounds on sample complexity for comparative learning, in terns of dependence on the error parameter $\epsilon$, the same as the gap in the corresponding bounds of Hu and Peale (2023)? It would appear that your lower bound scales with $\frac{1}{\epsilon}$ and your upper bound scales with $\frac{1}{\epsilon^2}$. The same can be observed in the bounds of Hu and Peale (2023). In that case, the sentence about this gap in your abstract, and similar later in the introduction, is misleading and potentially creating the wrong expectation, as in making believe that this gap is closed somehow by the work under review, which appears to not be the case, judging from Theorem 4.

- **Correctness:** I have a concern about this. For the setting called "general comparative learning" it appears that the lower bound (Theorem 13) scales with $\frac{1}{\epsilon^2}$ and the upper bound (Theorem 12) scales with $\frac{1}{\epsilon}\log(\frac{1}{\epsilon})$. Just from this one could conclude that there is something wrong with the arguments.

- **Suitability for ALT:** The problem setting appears to be of interest to the ALT readership.


As requested, here is a **summary of pros and cons:**

>**Pros:**
>- Mostly well-written paper, and content is suitable for ALT.
>- Addressing an interesting problem, and motivations and discussions are generally believable.

>**Cons:**
>- Unclarity about the scope of the contributions and novelties with respect to Hu and Peale (2023).
>- Inconsistency (?) between lower bound and upper bound for the general case (see "correctness" above).

Last, I share a list of **detailed comments:**
- Abstract and other places say "error parameter 1/ϵ" which looks strange. Error parameter is ϵ, I think. Perhaps this is meant to say something about the dependence of the bounds on 1/ϵ, either linear or quadratic.
- Introduction, line 4: "all possible data-generating distributions." (replace "processes" with "distributions")
- Four lines further down: delete "so called"
- Couple of lines down: "data-generating distribution" (replace "process" with "distribution")
- Next paragraph: After commenting on PAC bounds being vacuous for deep learning, perhaps add a comment that there is some recent literature on PAC-Bayes bounds giving non-vacuous values for deep learning such as "Computing nonvacuous generalization bounds for deep (stochastic) neural networks with many more parameters than training data" (Dziugaite and Roy, 2017) and "Tighter risk certificates for neural networks" (Pérez-Ortiz et al., 2021).
- Next paragraph: Maybe " label-separatedness" should be " label-conditional separatedness"?
- Next paragraph, first line: "the PAC learning framework"
- Next paragraph: The case of comparative learning that is proper w.r.t the benchmark class is clear (meaning the learner is required to output a predictor from the benchmark class). However, the last line of this paragraph says "For both general and benchmark-proper comparative learning" and it is not very clear at this point what defines the case of general comparative learning. (At the beginning of Section 4 this is clarified: general comparative learning means that no properness requirement is imposed. I suggest to make this clear a this point in the introduction.)
- Next page, second bullet: "as well as general benchmark-proper learning" - now this created a complete confusion since before there were two cases, one being the benchmark-proper comparative learning case, and the other one being the general comparative learning case. So, then what is this "general benchmark-proper learning" ?
- Section 2, first line: "standard statistical learning theory framework"
- Next paragraph: The note that "a partial classifier can be viewed as a function $h : \mathcal{Z} → \mathcal{Y}$, where $\mathcal{Z} ⊆ \mathcal{X}$ is a subset of the domain" needs to clarify that $h$ and $\mathcal{Z}$ are connected. In fact, $\mathcal{Z} = h^{-1}(\star)$. The point being that it needs to be clarified that $\mathcal{Z}$ is not the same for all partial classifiers.
- The names "partial class" and "total class" are creating more confusion than clarity, I think.
- Better to say "class of partial classifiers" than saying "partial class" (the latter evokes a subset of a function class).
- Next: Notation $S$ for dataset is clashing a bit with notation $\mathcal{S}$ for source hypothesis class. As in, the visual effect of seeing $S$ and $\mathcal{S}$ tends to suggest some connection between the denoted objects. Could this be avoided?
- Definition 1: There is inconsistency between the use of $n_{\mathcal{S},\mathcal{B}}(\epsilon, \delta)$ and $n(\epsilon, \delta)$.
- Also, I suggest the notation should be $n_{\mathcal{S},\mathcal{B}}(\epsilon, \delta)$ with $n$ and not the one with '$\mathrm{n}$' written currently in the notation $\mathrm{n}_{\mathcal{S},\mathcal{B}}(\epsilon, \delta)$.
- Same comment for the sample complexities with upper script 'gen' and 'prop' and 'ERM'
- Next page, in Observation 1: Maybe rewrite as "There exist classes $\mathcal{S}$ and $\mathcal{B}$ of binary total functions such that" (if that is the intended meaning)
- Corollary 8: Reformulate similarly to previous bullet, to clarify the intended meaning.
- ...
- The conclusion was kind of lame. Could an effort be made to produce an informative conclusion, with a recap of the highlights of the work, discussion of scope and limitations, and the take-aways messages that readers should remember from this paper.

%%% === POST REBUTTAL

The rebuttal has addressed my main concern. I've increased my score.

**Paper Award:**

No

---

> ### Author Response · Authors · 2024-11-25
> **There is no inconsistency in our bounds! Let us clarify:**
>
> Regarding the potential inconsistency between our bounds raised in the review
> ============================================================
>
> This is a misunderstanding, there is no inconsistency between our upper and lower bounds in Section 4.1! Note that the conditions for Theorem 12 and Theorem 13 are different. The lower upper bound in Theorem 12 involves the **diameter d**, while the lower bound in Theorem 13 involves the **joint diameter d’**. Please note the distinction between these parameters (Definition 11). Inspecting their definition immediately shows that the **joint diameter is upper bounded by the diameter**, and we also state this explicitly right after Definition 11.
>
> Thus, there is no single hypothesis class that satisfies the conditions of both theorems. Some classes satisfy conditions of Theorem 12 and have the faster rate of $\frac{d}{\epsilon}\log(\frac{d}{\epsilon})$ and some classes satisfy conditions of Theorem 13 and their sample complexity is lower bounded by $\frac{1}{\epsilon^2}$. Some classes satisfy neither and their sample complexity remains open. We hope this answers your concerns.

---

> ### Author Response · Authors · 2024-11-25
> **Response to other points raised in the review**
>
> Answers to other points raised
> =======================
>
> > “For instance, if what this work calls "proper" comparative learning (where the learner is required to output a predictor from the benchmark class) is a novel setting here, which was not considered by Hu and Peale (2023), this needs to be highlighted prominently and so does the associated analysis.”
>
> It is correct that Hu and Peale (2023) do not distinguish between proper and improper learners. In fact, they use the algorithm for learning partial classes introduced in Alon et al (2021), which is generally an improper algorithm. We introduce the benchmark proper learning framework, which naturally models model distillation type learning scenarios. We highlight this under the first bullet points in our "Summary of Contributions" subsection. We will aim to highlight it more prominently throughout the introduction.
>
> Note that for the benchmark proper learning setting we further distinguish between benchmark-ERM learners and proper learners that are not ERMs (a proper learner does not necessarily output an empirical risk minimizing hypothesis from the class, thus benchmark ERM is a sub-case of benchmark proper learning). We will carefully inspect our manuscript for potential unclarities. But general learning versus proper learning versus ERM learning are also standard distinctions in the PAC learning literature.
>
>
> >  “Originality and significance: … . Also I'd like to ask the author(s) to address this question: Is not the gap between your upper and lower bounds on sample complexity for comparative learning, in terns of dependence on the error parameter , the same as the gap in the corresponding bounds of Hu and Peale (2023)? It would appear that your lower bound scales with $\frac{1}{\epsilon}$ and your upper bound scales with  $\frac{1}{\epsilon^2}$. The same can be observed in the bounds of Hu and Peale (2023). In that case, the sentence about this gap in your abstract, and similar later in the introduction, is misleading ..”
>
> We will inspect our introduction and abstract and aim to set correct expectations!
>
> We would like to note that our contributions are twofold: First, introduction and analysis of proper comparative learning (Section 3), and second, finding sufficient conditions for fast ($\frac{1}{\epsilon}$) and slow ($\frac{1}{\epsilon^2}$) rates for both proper comparative learning (Section 3) and general comparative learning (Section 4).  In Section 4.2 we provide general conditions on when we get these fast (Theorem 15) and slow rates (Theorem 17). This distinction between fast and slow rates was not made in Hu and Peale (2023).
>
> You are right that the conditions provided in our Theorems 15 and 17 are not complementary to each other and hence there could exist classes that do not fall under any of these two sets of conditions. So there still exist some gaps in the landscape, as we also acknowledge in our submission (see Section 5). Our results do settle the rate for other classes though.
>
> It is true that there is a $\frac{1}{\epsilon}$ gap in the result of Theorem 4, but the main point of this theorem and Section 3 in general is not to close the gap. The main goal there is to highlight the distinction between proper comparative learning and general comparative learning. We first show that the sample complexity of benchmark-ERM learners is controlled by a different dimension, the one-sided graph dimension, rather than the mutual VC dimension from the earlier work by Hu and Peale. Note that empirical risk minimization (ERM) is a very common learning approach, and we thus believe that this distinction provides relevant novel insights. Second, we show that there exist classes that are learnable by an improper learner but cannot be learned by any proper learner.
>
>  Lastly, we do discuss fast versus slow rates for general proper learners in the last part of Section 3, “Linear versus quadratic dependence on the error parameter”.
>
> Detailed comments
> ===============
>
> Thank you for your detailed comments! We will take all of them into consideration when updating our manuscript. We respond to two specifically:
>
> > “Next page, second bullet: "as well as general benchmark-proper learning" - now this created a complete confusion since before there were two cases, one being the benchmark-proper comparative learning case, and the other one being the general comparative learning case. So, then what is this "general benchmark-proper learning" ?”
>
> See our elaborations on this issue above. In this instance, we used the term “general benchmark-proper learning” to emphasize the distinction from ERM based proper learning. We can see how this created a potential for confusion and rephrase it. As noted above, we will inspect our manuscript and ensure non-ambiguous terminology throughout.
>
> > “..recent literature on PAC-Bayes bounds giving non-vacuous values”
>
> Thanks for pointing out the additional references! We will add and discuss them when updating our manuscript.

---

### Official Review · Reviewer_1CHX · 2024-11-06
**Solid paper, interesting contribution, and sound proof technique.**

**Rating:** 7
**Confidence:** 2

**Review:**

The paper studies the problem of comparative learning, a variant of PAC learning that allows labelling to come from one hypothesis class while the learner’s performance is measured against another (possibly different) hypothesis class. The authors derive precise conditions under which the learner performance is linear or quadratic in $1/\epsilon$.

Quality:
- The proof techniques and arguments are sound. The bounds on linear and quadratic dependence on 1/\epsilon rely on established techniques and are derived by relating the comparative learning framework to that of agnostic PAC learning under deterministic labels.

Clarity:

- The paper is mostly clear, and I did not spot a missing definition. There are many definitions of properties throughout the work, such as one-sided mutual graph dimension, which could benefit from being simplified/joined with other definitions for ease of reading. However, I understand that this is not always possible and it seems that these precise definitions are necessary for the results.

Originality:
- The paper seems to continue along the lines of the initial paper (Hu and Peale (2023)) which, to the best of my knowledge, proposes this comparative learning framework. Nevertheless, this paper is distinct from previous papers in that it precisely characterizes when linear vs. quadratic dependence in 1/\epsilon arises.

Significance:
- I find the results in the paper to be significant for learning theory and more broadly for the communities of model distillation. It is interesting and novel, at least from my perspective, to see learning bounds that are not majorly governed by the standard VC dimension.

Recommendations and Questions:
- Typos: “the the” bottom of page 8. “of learning for learning” top of page 5. n \geq n_{S, B}(\epsilon, \delta) in Definition 1. “both both”, middle of page 3. “have been shown in for”, middle of page 3.

**Paper Award:**

Yes

---

> ### Author Response · Authors · 2024-11-25
> **Thank you for your appreciation of our work!**
>
> We will fix the typos you pointed out!

---

### Official Review · Reviewer_Eicz · 2024-11-11
**Review of "Refining the Sample Complexity of Comparative Learning"**

**Rating:** 7
**Confidence:** 3

**Review:**

This paper studies the PAC sample complexity of comparative learning for both the benchmark proper and the general learning settings. For the benchmark proper setting, a new dimension (one-sided mutual graph dimension) is proposed based on which novel upper and lower bounds for the sample complexity of ERMs are proved. The authors also provide a family of examples which have mutual VC dimension 0 and one-sided mutual graph dimension increasing to infinity. They show that their lower bounds for ERMs also holds for any proper learner in those examples. They also propose a sufficient condition for quadratic dependence of sample complexity on $1/\epsilon$ in benchmark-proper setting. For the general comparative learning setting, they define diameter and joint diameter for partial classes via which sufficient conditions for both linear and quadratic dependence of sample complexity on $1/\epsilon$ are proposed.

Pros: The problem studied in this paper represents a meaningful and intriguing extension of standard PAC learning, offering substantial new avenues for exploration. The contribution of this paper is original and significant especially for the benchmark proper setting. The paper is well-organized and written clearly with substantial technical illustration.

Cons: I didn't find any major issues with the paper. I have the following comments for potential improvement.
1) For the proof of Theorem 12, there is no justification of the claim that "A sample of size
... contains every point with mass at least $\epsilon/d$ with probability more than $1-\delta$". I suggest the authors to add a reference or provide a justification for this claim.
2) In page 12, the following existing result is cited: "any agnostic learner for the agreement class comparatively learns source and benchmark in the sense of Definition 1 (Hu and Peale, 2023)". Since it is a key argument used in the proof of Theorem 15, I believe it would be better to cite it as a lemma or refer to the exact position where it is stated in Hu and Peale (2023).
3) There are some typos. In page 3, there are two "both" in the sentence "both both linear and quadratic dependence..." and there is no period in that sentence. In page 10 after the title of section 4, it should be $\tilde\Theta$ instead of $\tilde\theta$.

**Paper Award:**

No

---

> ### Author Response · Authors · 2024-11-25
> **Thank you for your appreciation of our work! Our answer the three points you raised:**
>
> 1. Thanks! This is a standard argument, but we will add this to the proof. For completeness we also state it here. Probability of a point with mass at least $\frac{\epsilon}{d}$ not appearing in a sample of size $m$ is upper bounded by $(1-\frac{\epsilon}{d})^m \le \exp(-m\frac{\epsilon}{d})$. There are at most $\frac{d}{\epsilon}$ such points, thus, probability that at least one of them does not appear in the sample is upper bounded by $\frac{d}{\epsilon}exp(-m\frac{\epsilon}{d})$ by a union bound. Setting this to be less than $\delta$ gives the desired results.
> 2. Thanks! We will add the precise citation: Hu and Peale (2023), Lemma 8, Page 72:18.
>
> 3. Thanks! We will fix these typos.

---

### Official Review · Reviewer_LcZ6 · 2024-11-11
**Some new results in a recently introduced variation of PAC-learning. Proofs are simple and similar to proofs of already existing results.**

**Rating:** 6
**Confidence:** 4

**Review:**

Comparative learning is a recently introduced variation of the PAC-framework.
In comparative learning, the labeling is assumed to be realizable by one
hypothesis class (the source class $S$), while the learner's performance
is measured against the best hypothesis of another class (the benchmark
class $B$). It had been shown by Hu and Peale that the comparative
learnability of $(S,B)$ can be characterized by the so-called mutual
VC-dimension of $(S,B)$. The latter is denoted by $\vc(S,B)$ and
the sample complexity grows linearly with this parameter. In this submission,
the authors consider benchmark-proper learners (whose final hypothesis
must be taken from $B$) or, even more restrictive, benchmark-ERM learners
(whose final hypothesis is taken from $B$ and must be one of the hypotheses
with minimum empirical error). The main results are as follows:

1.
A new combinatorial parameter $d_G^\rightarrow(S,B)$, called the one-sided mutual
graph dimension of $S$ and $B$, is introduced. It is shown that this
parameter characterizes the learnability with benchmark-ERM
learners. The sample complexity grows linearly with $d_G^\rightarrow(S,B)$.
It furthermore grows at least linearly, and at most quadratically,
with $1/\varepsilon$ where $\varepsilon$, as usual, denotes the accuracy parameter.

2.
The sample size required for comparative learning is denoted in the paper
by $n_{S,B}^{gen}(\varepsilon,\delta)$, respectively by $n_{S,B}^{prop}(\varepsilon,\delta)$
or by $n_{S,B}^{ERM}(\varepsilon,\delta)$ if the learner is assumed to be
benchmark-proper or benchmark-ERM, respectively. Clearly
\[
\vc(S,B) \le d_G^\rightarrow(S,B)\ \mbox{ and }\
n_{S,B}^{gen}(\varepsilon,\delta) \le n_{S,B}^{prop}(\varepsilon,\delta) \le
n_{S,B}^{ERM}(\varepsilon,\delta) \enspace .
\]
The authors present an example which demonstrates that the gap between
the parameters $\vc(S,B)$ and $d_G^\rightarrow(S,B)$ can be arbitrarily large.
An analogous remark applies to the
parameters $n_{S,B}^{gen}$, $n_{S,B}^{prop}$ and $ n_{S,B}^{ERM}$.
This implies that there exists a pair $(S,B)$ that is comparatively learnable
but not by a benchmark-proper learner. And there exists a pair $(S,B)$
that is comparatively learnable by a benchmark-proper learner but not
by means of benchmark-ERM.

3.
The authors examine more closely how the sample complexity may depend on $\varepsilon$.
They remind the reader of special extreme forms of comparative learning
(like, for instance,  classical PAC-learning in the realizable case
or properly learning a known true labeling) which demonstrate that both,
linear and quadratic dependence, may actually happen. They furthermore
present a variety of bounds on the sample complexity (in terms of
dimensions different from $d_G^\rightarrow$) which shed some light on the
question under which conditions we may expect a quadratic, respectively
sub-quadratic, dependence on $1/\varepsilon$.

4.
In the appendix, the authors introduce the mutual graph dimension $d_G$.
It is shown that, if $d:= d_G(S,B) <\infty$, then $(S,B)$ can be
comparatively learned using either source-ERM or benchmark-ERM.
Moreover the sample complexity of these learners is upper bounded
by $O\left(\frac{1}{\varepsilon^2}(d + \log(1/\delta))\right)$.


The authors make a reasonable contribution within the relatively new
framework of comparative learning. They extend the analysis of general
comparative learners in (Hu and Peale, 2023) by an analysis of
benchmark-proper and benchmark-ERM learners. As far as I can see, the
results and the proofs in the paper are correct.

On the other hand, the proofs in the paper are relatively easy to obtain.
Most of them are close relatives of existing proofs in the standard
PAC-learning framework. The proof of Lemma 6 looks more complicated but,
I think, it can be considerably simplified. See the technical comments
below.

Summary:
This is a nice contribution to a relatively new learning framework
but, as far as I can see, there were no major technical hurdles to
overcome.

\centerline{\bf Technical Comments}

Definition 1:
$n(\varepsilon,\delta)$ should be replaced by $n_{S,B}(\varepsilon,\delta)$.

Proof of Observation 1:
It should be said explicitly that $X$ (which is not specified further
in the proof) is an infinite set.

Theorem 4:
$\tilde O$ should be replaced by $O$ in this Theorem. Compare with Lemma 6.

Proof of Theorem 15:
In the proof, you run three learners. Wouldn't it be simpler to run the
learner whose associated combinatorial parameter is the smallest one
(Learner 1 if $d = \vc(S)$, Learner 2 if $d = \diam(B)$,
Learner 3 if $d = \diam(A_{S,B})$)?

Proof of Lemma 6:
I think, the proof can be given in a very simple manner.
The inequality (1) in Lemma 27 is verified in the same way as it is
done in (Anthony and Bartlett, 2002). For $b \in B$, define
\[
R(b) = \{(S,T): |L_S(b) - L_T(b)| > \varepsilon/2\}
\]
so that $R(B) = \cup_{b \in B}R(b)$. For each $b \in B$, the
term $\Pr_\sigma[\sigma(z) \in R(b)]$ can be bounded from above by means
of Hoeffding's inequality (as it is done in (Anthony and Bartlett, 2002)).
In order to bound $\Pr_\sigma[\sigma(z) \in R(B)]$ from above, one plans
to apply the union bound. A naive application would introduce the
factor $|B|$. The crucial observation is that $R(b)$ depends on $b$
only weakly:

a)
Let $z = (z_1,\ldots,z_{2m})$ and $z_i = (x_i,y_i)$. With each $b \in B$
associate the zero-one loss function
\[
h_b(z_i) = \left\{ \begin{array}{ll}
   0 & \mbox{if $b(x_i) = y_i$} \\
   1 & \mbox{if $b(x_i) \neq y_i$}
           \end{array} \right. \enspace .
\]

b)
Observe that $h_b = h_{b'}$ implies that $R(b) = R(b')$.

Thus, an application of the union bound yields only
factor $s := |\{h_b: b \in B\}|$ instead of factor $|B|$.
Let $d := d_G^\rightarrow(S,B)$ and note that the VC-dimension of
the class $\{h_b: b \in B\}$ cannot exceed $d$. An application
of Sauer's bound yields $s \le \left(\frac{2em}{d}\right)^d$.
Now the proof can be completed as in (Anthony and Bartlett, 2002). \\
Actually the whole proof is almost the same as in (Anthony and Bartlett, 2002).
The only thing is that one has to bring the hypothesis
class $\{h_b: b \in B\}$ into play and one has to observe
that $\vc(\{h_b: b \in B\}) \le d_G^\rightarrow(S,B)$. \\
Note that the functions $h_b$ are 0,1-valued. There is no need to make
a detour on partially defined hypothesis classes. Lemma 28 is
therefore not needed here.

Theorem 32:
Does the logical ``or'' in ``benchmark-properly or source-properly''
express what you mean? I guess, you mean that both forms of proper
comparative learning have a sample complexity upper bound as given
in the theorem.

Proof of Theorem 32:
Which hypothesis do you denote by $\bar a_{s,b}$ in this proof.
Do you mean $a_{s,b}$? \\
Replace $((X \times Y \times ) \{0\})^m$ by $((X \times Y) \times \{0\})^m$.

Proof of Corollary 33:
Shouldn't the expression $\frac{1}{n} \sum_{i=1}^{n} 1[a_{s,b}(x,y) \neq 1]$
represent the empirical error of $a_{s,b}$? Then this expression should look
like $\frac{1}{n} \sum_{i=1}^{n} 1[a_{s,b}(x_i,y_i) \neq 0]$.


General Remark:
In several definitions of sets in the paper, one finds unnecessary occurencies
of ``$\forall$''. For instance in Lemma 31 (but not only there),
the set $(U,g(U))$ is defined as $\{(x,g(x)): \forall x \in U\}$.
The symbol ``$\forall$'' should be removed. Please check the whole paper
for meaningless occurencies of ``$\forall$''.

**Paper Award:**

No

---

> ### Author Response · Authors · 2024-11-25
> **Thank you for your careful reading of our work!**
>
> >“Proof of Observation 1”
>
> Thanks! We will clearly state this in the final version of the paper.
>
> >“Proof of Theorem 15: In the proof, you run three learners. Wouldn't it be simpler to run the learner whose associated combinatorial parameter is the smallest one”:
>
> You are correct in the case where the learner has access to the three parameters (or for cases where it would be straight-forward to compute them for the involved classes). The algorithm and analysis we suggest here is slightly more general in that the learner itself is independent of the three parameters. It is only the analysis of the sample complexity that depends on them. For our learner, the sample complexity scales with the minimum of the three parameters without the learner needing the parameters as an input. We will add a comment to discuss this distinction and also add that in the case where the parameters are known to the learner, running one of the three “sub-learners” is sufficient.
>
> >“Proof of Lemma 6:...”
>
> Thank you for suggesting this simpler proof! We agree that there are multiple ways to prove this result and involving partial classes is not required. This also follows from the line of argument we sketch in the main part of our submission. We will clarify this when updating our work.
>
> >“Theorem 32…”
>
> Yes, both forms of benchmark and source proper learners have the same stated upper bound. We will clarify this, thank you for pointing it out!
>
> >“Definition 1…”, “Theorem 4…”, “Proof of Theorem 32…”, and “Proof of Corollary 33…”
>
> Thanks, these are all indeed typos. We will fix them.
>
> >“General Remark…”
>
> Thank you for spotting this. We will fix all such occurrences.

---

> > ### Comment · Reviewer_LcZ6 · 2024-11-25
> > **Proof of Theorem 15**
> >
> > As for proving Theorem 15, it is clearly allowed to use the learner with the smallest associated combinatorial parameter because the learner may be specialized to the classes S and B. The issue that you raise in your comment is how we know (or how a meta-algorithm knows) which of the three potential algorithms is the best one. This issue is irrelevant for proving Theorem 15.

---

> > > ### Author Response · Authors · 2024-11-29
> > > **Agreed.**
> > >
> > > Yes, you are right that for the sample complexity statement in Theorem 15, this distinction is not required. We will take it out of the proof of Theorem 15, and instead add a comment to discuss the distinction. Ie add a comment that says there exists one learner that automatically achieves the sample complexity stated in the theorem for all pairs of classes (together with the argument that's currently given in the proof).

---

> > > > ### Comment · Reviewer_LcZ6 · 2024-11-30
> > > >
> > > > That's fine.

---

### Author Rebuttal · Authors · 2024-11-25

We thank all reviewers for their careful reading and appreciation of our work!

We appreciate the suggested edits, minor corrections, typos etc and will take them into careful consideration when updating our manuscript.

We respond to specific issues raised by reviewers in individual responses below.

---

### Meta-Review · Area_Chair_SQQt · 2024-12-07

**Recommendation:** Accept
**Confidence:** 4

**Metareview:**

Comparative learning is a recently introduced variation of the PAC-framework.
It had been shown by Hu and Peale, ICTS 2023, that the comparative learnability  of a pair $(S,B)$ of hypotheses classes can be characterized by the so-called
mutual VC-dimension of $(S,B)$. In this submission, the authors consider
benchmark-proper learners (whose final hypothesis must be taken from the
benchmark class $B$) or, even more restrictive, benchmark-ERM learners
(whose final hypothesis must be an ERM-hypothesis taken from $B$).
It is shown that benchmark-ERM learnability of $(S,B)$ can be characterized
by a new combinatorial parameter named the one-sided mutual graph dimension
of $(S,B)$. It is furthermore shown that, in a strong sense, arbitrary learners
in the comparative setting are much more powerful than their proper counterparts,
and the latter are much more powerful than benchmark-ERM learners. Last not least,
the authors take a closer look on how the diverse sample size bounds depend on
the accuracy parameter $\epsilon$ and under which conditions sample size grows
superlinearly in dependence of $1/\epsilon$.

Pros:
This is a nice contribution to the relatively new framework of comparative
learning. The authors extend the analysis of general comparative learners
in (Hu and Peale, ICTS 2023) by an analysis of benchmark-proper and
benchmark-ERM learners. The results and the proofs in the paper are correct.
(One of the reviewers raised some doubts about the correctness but these doubts
are completely unfounded.)

Cons:
The proofs in the paper are relatively easy to obtain.
Most of them are close relatives of existing proofs in the standard
PAC-learning framework. The only proof that seems to be more involved
is the proof of Lemma 6. However, as explained in detail in one of the
reviews, this proof can be considerably simplified (and then also becomes
a close relative of an already existing proof for a classical result in
statistical learning theory).

Summary:
Despite of the relative simplicity of the proofs, I think this is a nice
extension of the last year's work of Hu and Peale. And the topic seems  to fit very well with the ALT-Conference. For this reason, I would be
in favor of acceptance.

Comment:
 I am quite confident that this would be a nice paper for ALT but, because of the relative simplicity of the proofs, I am not absolutely certain. It is imaginable that there are many competing papers that, on top of fitting nicely with the ALT-conference as well, are technically stronger.

**Paper Award:**

No